# MiClip: Learning to Interpret Representation in Vision Models

**Yingdong Shi**[*], **Zhiyu Yang**[*], **Changming Li, Jingyi Yu, Kan Ren**[†]
ShanghaiTech University
`{shiyd2023, yangzhy22022, renkan}@shanghaitech.edu.cn`

## Abstract

Vision models have demonstrated remarkable capabilities, yet their decision-making processes remain largely opaque. Mechanistic interpretability (MI) offers a promising avenue to decode these internal workings. However, existing interpretation methods suffer from two key limitations. First, they rely on the flawed *activation-magnitude assumption*, assuming that the importance of a neuron is directly reflected by the magnitude of its activation, which ignores more nuanced causal roles. Second, they are predominantly *input-centric*, failing to capture the causal mechanisms that drive a model's output. These shortcomings lead to inaccurate and unreliable internal representation interpretations, especially in cases of incorrect predictions. We propose MiClip (Mechanism-Interpretability via Contrastive Learning), a novel framework that extends CLIP's contrastive learning to align internal mechanisms of vision models with general semantic concepts, enabling interpretable and controllable representations. Our approach circumvents previous limitations by performing multimodal alignment between a model's internal representations and both its input concepts and output semantics via contrastive learning. We demonstrate that MiClip is a general framework applicable to diverse representation unit types, including individual neurons and sparse autoencoder (SAE) features. By enabling precise, causal-aware interpretation, MiClip not only reveals the semantic properties of a model's internals but also paves the way for effective and targeted manipulation of model behaviors. Our project homepage is available at `https://foundation-model-research.github.io/MICLIP`.

## 1 Introduction

*Mechanistic interpretability* (MI) (Zeiler & Fergus, 2014; Oikarinen & Weng, 2023) on vision models (Dosovitskiy et al., 2021; Rombach et al., 2022) offers a promising avenue for making models more transparent and controllable. MI aims to uncover models' internal mechanisms, *e.g.*, hidden representations and computational circuits, and relate them to both model behavior and human-understandable concepts. For instance, Wang et al. (2025) shows that tracing the successive transformation of inputs into outputs can reveal how object concepts are processed within vision models. Furthermore, once such mechanisms are identified, they can be systematically manipulated to steer models toward desired behaviors (Li et al., 2024; Ferrando et al., 2025; Shi et al., 2025).

Researchers have proposed a range of interpretability methodologies to decode the internal representations of vision models into human-understandable semantics (Olah et al., 2017). Early efforts largely relied on manual annotation and feature visualization (Zeiler & Fergus, 2014; Selvaraju et al., 2017). More recent work instead analyzes units within the hidden representations, such as individual neurons, and associates them with semantic concepts by measuring the co-occurrence between high neuron activations and the presence of corresponding concepts in input images (Bau et al., 2017; Oikarinen & Weng, 2023; Zhang et al., 2024; Bai et al., 2025).

Despite these advances, existing approaches face notable limitations. First, most prior work relies on the *activation-magnitude assumption*: for any representation unit, a larger activation value is inter-

---

[*]Equal contribution.
[†]Correspondence to Kan Ren.

preted as indicating a stronger presence of the unit's associated concept in the model's information-processing pipeline. However, a neuron's contribution to model behavior is often more complex. An increase in activation value does not necessarily imply the occurrence of the corresponding concept during inference. Conversely, even negative activations can positively influence the model's prediction of certain concepts. Second, existing methods are predominantly *input-centric*, focusing on aligning internal representations with concepts present in the input. This paradigm introduces several issues, as it is not grounded in the causal mechanisms that actually drive model behavior (Gur-Arieh et al., 2025). Such limitations are particularly evident when the model produces incorrect predictions. Similarly, input-centric methods (Bau et al., 2017; Zhang et al., 2024; Bai et al., 2025) fail in cases of incorrect predictions, as they do not capture the intrinsic causal mechanisms underlying the model's decision-making.

In this paper, we introduce MiCLIP, an MI framework for vision models from a novel functionality perspective on the model's representation units[1]. Rather than relying on the conventional activation-magnitude assumption, MiCLIP represents a target unit, such as an individual neuron or a sparse feature (Huben et al., 2024), as a semantic vector embedded within a human-understandable space, such as the CLIP (Radford et al., 2021) semantic space. By directly measuring the semantic relatedness of internal units to specific concepts, our approach bypasses the limitations of activation-magnitude-based methods.

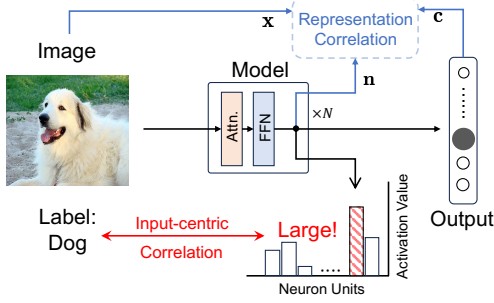

Figure 1: Traditional input-centric methods identify neurons by correlating their activation magnitudes with the given input label (*e.g.*, "Dog"). In contrast, our framework instead establishes correlations in the representation space and takes both the input and the output into consideration.

To achieve this, MiCLIP performs multimodal alignment between hidden representation units and concepts derived from both inputs and outputs of the model. In particular, contrastive learning (Radford et al., 2021) is employed to ground feature functionality with respect to input concepts and output semantics. This dual grounding integrates input- and output-centric perspectives, thereby revealing the causal trajectory of information processing: from input, through internal units, to model outputs as shown in Figure 1. In doing so, MiCLIP not only provides more faithful interpretability of the mechanisms underlying model behavior, but also enables direct model steering by manipulating concept-aligned internal units.

Our work advances MI study in vision models with key contributions listed as follows: (i) To the best of our knowledge, this is the first study on learning semantic representations of the model internal features and aligning them with human-understandable semantic spaces. This multimodal grounding provides a unified and generalizable interpretability framework across diverse vision architectures. (ii) Unlike prior input-centric explanation or attribution methods, MiCLIP incorporates both input- and output-grounded semantics, aligning the entire reasoning trajectory of the model with human concepts. (iii) MiCLIP applies broadly to different forms of internal representation units, including individual neurons and SAE features (Huben et al., 2024; Gao et al., 2025), offering a versatile tool for understanding and steering vision models.

## 2 BACKGROUND AND RELATED WORK

Existing approaches that aim to associate textual concepts with internal representation units in vision models remain largely input-centric and can be grouped into two main trends.

**Activation-based.** The first line of work selects inputs that strongly activate a neuron or crops out the highly activating region (Kalibhat et al., 2023) and then interprets the recurring patterns. This can be done by correlating activations with annotated concepts (Bau et al., 2017), or by leveraging pretrained models to automatically generate textual descriptions for the highly activating inputs (Hernandez et al., 2021; Kalibhat et al., 2023; Zhang et al., 2024; Bai et al., 2025). Network

---

[1]Representation units include neurons (dimension of the representation) and features (direction in the activation space (Huben et al., 2024)) that can be learned by sparse autoencoders (Gao et al., 2025).

Dissection (Bau et al., 2017) quantifies the correlation between hidden units and concepts by computing the intersection-over-union (IoU) between thresholded activation maps and pixel-level annotations in the Broden dataset. Describe-and-Dissect (DnD) (Oikarinen & Weng, 2023) follows the same activation-selection paradigm, identifying top activating images for each neuron and then assigning textual concepts by matching them with vision-language embeddings. Similarly, automated framework V-Interp (Zhang et al., 2024) relies on highly activating samples as prompts to large multimodal models, which then produce free-form textual explanations of neurons and features. Despite their advances, these methods all rely on the *activation-magnitude assumption*, presuming that larger activations correspond to a stronger presence of a concept, which does not necessarily hold and fails to capture a unit's causal influence on the model's output.

**Representation-based.** The second trend constructs representations for both model internals and semantic concepts and aligns them to caption (Oikarinen & Weng, 2023; Balasubramanian et al., 2024). Balasubramanian et al. (2024) aims at explaining the roles of ViT components (*e.g.* attention head) by decomposing the contribution vector in the final layer's activation, then learns a set of linear maps for each component to align them in CLIP's embedding space, enabling text description. However, this method is restricted to ViT submodules and does not generalize to more fine-grained units such as neurons or features at arbitrary positions in diverse vision models.

CLIP-Dissect (Oikarinen & Weng, 2023) prepares a probing dataset $\mathcal{D}$ and a concept set $\mathcal{C}$. For each neuron $k$, it constructs an activation vector $\mathrm{Act}(k, \mathcal{D}) \in \mathbb{R}^{|\mathcal{D}|}$ by recording its responses across all samples in $\mathcal{D}$ as the neuron representation. For each concept $c \in \mathcal{C}$, it builds a similarity profile $\mathrm{sim}(c, \mathcal{D})$ using CLIP embeddings of text and images as the concept representation. The most correlated concept for neuron $k$ is then identified as Equation 1, where $\mathbf{F}(\cdot)$ is a handcrafted similarity.

$$\arg\max_{c \in \mathcal{C}} \ \mathbf{F}\left(\mathrm{Act}(k, \mathcal{D}), \mathrm{sim}(c, \mathcal{D})\right). \tag{1}$$

Despite its strong performance, it relies on heuristically constructed neuron representations, which are still grounded in the *activation-magnitude assumption*. In contrast, MICLIP learns representations for model internals directly through contrastive training, avoiding heuristic designs and enabling more principled and flexible representation-concept alignment across diverse vision models.

**Limitations of input-centric interpretability.** A majority of existing works on explaining model internals with concepts remains input-centric: they infer a unit's meaning from correlations between its high activations and input-side patterns or captions alone (Bau et al., 2017; Hernandez et al., 2021; Zhang et al., 2024; Bai et al., 2025; Oikarinen & Weng, 2023). This paradigm implicitly assumes the *activation-magnitude assumption* yet provides no guarantee that the unit *causally* steers the model's predictions in that concept's direction. Recent work formalizes these concerns from an output-centric perspective, emphasizing that choosing the right causal mediator is central to faithful explanations (Mueller et al., 2024), and demonstrates in language models that descriptions grounded to both input and output yield more behaviorally faithful characterizations than input correlations alone (Gur-Arieh et al., 2025). Parallel evidence in vision (Gandelsman et al., 2025) proposes a CLIP-specific, output-centric method that interprets neurons by tracing their second-order effects on the model's output embeddings, thereby revealing each unit's causal semantic influence rather than just its activation correlations. Inspired by this discussion, MICLIP serves as a universal framework that aligns internal representations with human-understandable concepts retrieved from both input image semantics and the model's output decision.

## 3 METHODOLOGY

In this section, we introduce our proposed method MICLIP in detail, including how it adopts the contrastive learning paradigm to create a learned representation for model internals (Section 3.1), how it utilizes the representation to connect model internals with human-understandable semantics, enabling precise and interpretable descriptions of model internals, as well as accurate identification of concept-relevant components in the model (Section 3.2). In Section 3.3, we will discuss how MICLIP supports fine-grained model steering through unit-level interventions based on the identified representation units. The overall framework is demonstrated in Figure 2.

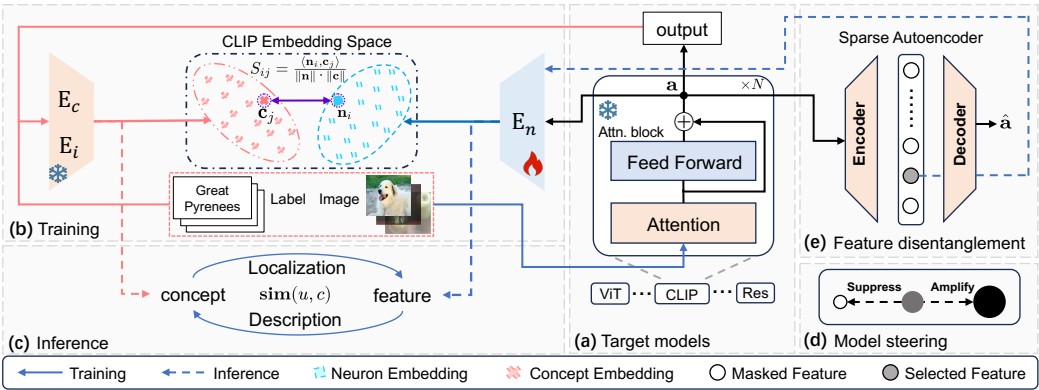

Figure 2: Framework of MiCLIP on (a) the target models (frozen), (b) contrastive learning of the shared embedding space (Section 3.1), (c) mechanistic feature localization and description (Section 3.2), (d) model steering (Section 3.3), (e) feature disentanglement with k-SAE (Optional).

## 3.1 MECHANISM-CONCEPT ALIGNMENT VIA CONTRASTIVE LEARNING

The CLIP framework (Radford et al., 2021) is pretrained to associate visual and textual modalities within a unified embedding space. Inspired by this, we extend CLIP's contrastive formulation to directly learn mappings from the activation space into CLIP's embedding space. This learned alignment avoids heuristic design and enables an interpretation of representation units in terms of human-understandable concepts, grounded on both inputs and outputs.

Given a labeled image dataset $\mathcal{D} = \{(x_i, c_i)\}_{i=1}^N$ with input images $x_i \in \mathcal{X}$, we forward them through the target vision model to retrieve activations $\mathbf{a}_i \in \mathbb{R}^n$ from the residual stream of the target layer and the corresponding predicted labels $\hat{c}_i \in \mathcal{C}$. This yields two paired sets: activations $\mathcal{A} = \{\mathbf{a}_i\}_{i=1}^N$ and predicted concepts $\{\hat{c}_i\}_{i=1}^N$. Then, the mechanism–concept alignment process within MiCLIP is formulated as a CLIP-based contrastive loss, specifically the symmetric InfoNCE (He et al., 2020) loss, including alignment between neurons (or features) and concepts, as well as between neurons and input images. More details are shown in Appendix B.1.

$$\mathcal{L}_{\text{alignment}} = \underbrace{\mathcal{L}_{\text{CLIP}}^{\text{out}} \left( \mathrm{E}_{\mathrm{n}}(\mathcal{A}; \theta_{\mathrm{n}}), \mathrm{E}_{\mathrm{c}}(\{\hat{c}_i\}_{i=1}^N) \right)}_{\text{neuron-concept loss}} + \underbrace{\mathcal{L}_{\text{CLIP}}^{\text{in}} \left( \mathrm{E}_{\mathrm{n}}(\mathcal{A}; \theta_{\mathrm{n}}), \mathrm{E}_{\mathrm{i}}(\mathcal{X}) \right)}_{\text{neuron-image loss}} \tag{2}$$

Here, the projection functions are implemented as CLIP-based encoders, which include a trainable neuron encoder $\mathrm{E}_{\mathrm{n}}(\cdot; \theta_{\mathrm{n}})$ parameterized by $\theta_{\mathrm{n}}$ and frozen encoders from a previously trained CLIP model: a concept encoder $\mathrm{E}_{\mathrm{c}}(\cdot)$ and an image encoder $\mathrm{E}_{\mathrm{i}}(\cdot)$. Specifically, $\mathrm{E}_{\mathrm{n}}$ maps the original neuron representation $\mathbf{a} \in \mathbb{R}^n$ to a neuron embedding $\mathbf{n} = \mathrm{E}_{\mathrm{n}}(\mathbf{a}) \in \mathbb{R}^d$, $\mathrm{E}_{\mathrm{c}}$ maps the concept $c \in \mathcal{C}$ to a concept embedding $\mathbf{c} = \mathrm{E}_{\mathrm{c}}(c) \in \mathbb{R}^d$, and $\mathrm{E}_{\mathrm{i}}$ maps the image $x \in \mathcal{X}$ to an image embedding $\mathbf{x} = \mathrm{E}_{\mathrm{i}}(x) \in \mathbb{R}^d$. Here, the neuron encoder $\mathrm{E}_{\mathrm{n}}(\cdot; \theta_{\mathrm{n}})$ connects the internal mechanism of the target model to the comprehensive semantic space of CLIP, which enables mechanistic feature localization and description, as discussed in Section 3.2.

**Discussion.** Compared to prior works (Kalibhat et al., 2023; Hernandez et al., 2021; Oikarinen & Weng, 2023; Balasubramanian et al., 2024) discussed in Section 2, MiCLIP introduces several key advantages through its contrastive learning paradigm. For instance, CLIP-Dissect constructs neuron–concept alignments heuristically through probing dataset correlations, whereas our approach replaces such heuristics with a learning-based mapping into the semantic space. On the other hand, Balasubramanian et al. (2024) emphasizes module-level decomposition of ViTs rather than fine-grained internal units. By contrast, MiCLIP provides a general, principled, and learning-based alignment framework for diverse internal representations.

## 3.2 MECHANISM LOCALIZATION AND DESCRIPTION

Once trained, MiCLIP enables both concept-to-mechanism localization and mechanism-to-concept description. This is made possible by encoding both the target model's representation and concept

spaces into a shared embedding space, where symmetric identification becomes feasible. This section describes how operations within MICLIP's unified embedding space support these two tasks.

Our core idea is to characterize the relationship between a representation unit of the target model and a human-understandable concept by comparing the relevance score $\mathbf{sim}(\cdot, \cdot)$ between their embeddings in the unified semantic space of MICLIP. Here, the representation unit can be either neurons or features learned from SAE, making the framework more generalizable.

To obtain the embedding for a representation unit, we adopt the encoder $\mathrm{E_n}(\cdot)$ to project it into MICLIP's embedding space. If the representation unit $u$ is a specific neuron with activation value $a_i \in \mathbb{R}$, it is directly encoded as $\mathbf{u} = \mathrm{E_n}(a_i \cdot e^{(i)})$ if it represents dimension $i$, where $e^{(i)} \in \mathbb{R}^n$ is the standard basis vector with 1 at position $i$. If $u$ is a SAE feature $f_i \in \mathbb{R}^n$ from the encoder dictionary, the embedding is obtained by $\mathbf{u} = \mathrm{E_n}(f_i)$. Meanwhile, the embedding of concept $c$ is $\mathbf{c} = \mathrm{E_c}(c)$. The relevance score $\mathbf{sim}(u, c)$ between a mechanism $u$ and a concept $c$ is then defined as the cosine similarity $\mathbf{sim}(u, c) = \frac{\mathbf{u} \cdot \mathbf{c}}{\|\mathbf{u}\| \cdot \|\mathbf{c}\|}$ of their embeddings $\mathbf{u}$ and $\mathbf{c}$. More information about SAE is mentioned in Appendix C.1. Notably, the linear design of $\mathrm{E_n}$ in Section 4.1 ensures consistent localization of individual neurons with the training process on full activation vectors. A detailed theoretical derivation of this consistency is provided in Appendix B.3.

With this similarity measure, we can find the most relevant representation units or concepts given their counterparts.

**Concept-to-Mechanism Localization.** Given a human-interpretable semantic concept $c \in \mathcal{C}$, this task aims to identify the mechanisms from a set of all representation units $\mathcal{U}$ (*e.g.*, all neurons or all interpretable features) that closely align with $c$. We identify the top $\tau$ representation units related to a concept $c$ and record their indices according to their similarity score in the set $\mathbf{L}_c$:

$$\mathbf{L}_c = \underset{i}{\text{SelectTop-}\tau} \left( \{ \mathbf{sim}(u_i, c) \}_{u_i \in \mathcal{U}} \right), \tag{3}$$

where SelectTop-$\tau$ selects the indices of the top $T$ elements. This process identifies the most influential mechanisms within the model that are responsible for a given concept.

**Mechanism-to-Concept Description.** Given a specific representation unit $u$, this task aims to find the concepts from a set $\mathcal{C}$ that best describe the mechanism. We identify the top $\tau$ concepts related to a representation unit $u$ and record the concepts according to their relevance score in the set $\mathbf{D}_u$:

$$\mathbf{D}_u = \underset{j}{\text{SelectTop-}\tau} \left( \{ \mathbf{sim}(u, c_j) \}_{c_j \in \mathcal{C}} \right). \tag{4}$$

### 3.3 MODEL CONTROL WITH MECHANISM INTERVENTION

To steer the target model, we intervene on the representation units identified in Section 3.2. For a given concept $c$, we collect its corresponding units indexed by $\mathbf{L}_c$ (neurons or SAE features), and adjust their activations to suppress or amplify the concept's influence on the model.

First, we apply an intervention to the target units. Then, these modified mechanisms are decoded into the original neuron space for subsequent operations. For each representation units $u$ ($u = a_i \cdot e^{(i)}$ or $u = f_i$) indexed within $\mathbf{L}_c$, we either apply a scalar multiplication or add an additive bias as

$$\tilde{u}_i = \beta u_i \ (\text{Scaling}) \quad \text{or} \quad \tilde{u}_i = u_i + \beta \ (\text{Adding}), \quad \forall i \in \mathbf{L}_c, \beta \in \mathbb{R}. \tag{5}$$

By applying a distinct parameter $\beta$, we can suppress or amplify the target feature to adjust the model.

In Section 4.2, we present intervention experiments that empirically demonstrate how targeted manipulations enable fine-grained model control, thereby validating the precision of concept-to-mechanism localization in MICLIP.

## 4 EXPERIMENT

### 4.1 EXPERIMENTAL SETTING

**Target models for interpretation.** We evaluate different methods on (i) image classification models trained on ImageNet-1k dataset (Deng et al., 2009) including, ResNet-50 (He et al., 2016) and ViT-B-16 (Dosovitskiy et al., 2021) (ii) pretrained multimodal models, specifically CLIP/ViT-B-16, for which we use its zero-shot classification ability to test the intervention in Section 4.3.

**Baselines.** We compare MICLIP against several baselines that are categorized into neuron-based and feature-based methods. Different baselines are selected for each experiment based on their applicability to the specific setting. We include Network Dissection (Bau et al., 2017), CLIP-Dissect (Oikarinen & Weng, 2023), V-Interp (Zhang et al., 2024) that relys on activation-magnitude assumption to identify concepts. Additionally, we construct a method named Act-Values, which identifies concepts according to the neuron activation values. More details are provided in Appendix D.1. For each of our baselines, we use 100,000 images randomly sampled from the ImageNet-1k (Russakovsky et al., 2015) training set for computation or training.

**Implementation details.** We train MICLIP on a subset of 100,000 images sampled from the ImageNet-1k training set. We use the frozen pretrained text encoder $E_c$ and vision encoder $E_i$ in CLIP/ViT-B-16, and we train a neuron encoder $E_n$ with a linear projection layer to map neuron activations to the embedding space. We use k-SAE (Gao et al., 2025) for feature disentangling, which is trained on the residual stream of a model. We validate our intervention approach on ImageNet-1k and further evaluate its generalization capabilities on an unseen dataset DTD (Cimpoi et al., 2014), which is a texture-based image classification dataset consisting of 56 texture categories.

## 4.2 EVALUATION ON MECHANISTIC INTERPRETATION AND LOCALIZATION

**Analysis 1: Quantitative Results on Mechanism Description.** We adopt the CLIP-Dissect evaluation framework, which benchmarks description accuracy on the final classification layer. In this setting, each neuron's *ground-truth* function is simply its corresponding class name (*e.g.*, "sea lion"), allowing for direct and objective evaluation. This setup enables an objective and scalable evaluation based on the neuron's actual function, avoiding the limitations and subjectivity of human assessments based on a few top-activating images.

We evaluate performance with three metrics. For the open-vocabulary experiments, we follow CLIP-Dissect and use the same open-ended concept sets of varying sizes (Common-3k, Common-10k, and Common-20k), which consist of the 3,000, 10,000, and 20,000 most common English words, respectively. For closed-set concept sets (1,000 ImageNet-1k classes), we **additionally** report **Accuracy (Acc.)**, the percentage of neurons whose top-ranked description exactly matches the *ground-truth* class. For general tasks, we use **CLIP Score** and **Mpnet Score**, which measure the cosine similarity between the generated description and the *ground-truth* class name using CLIP/ViT-B-16 and Mpnet-base-v2 encoders, respectively. Finally, to ensure the reliability of our improvements, we perform a **one-tailed paired t-test** across three random seeds to verify the statistical significance of our model against the baselines.

Table 1: Evaluation on neuron description. We compare with baseline methods on interpreted concepts for the neurons of the last layer in different vision models. ↑: higher is better. (* indicates statistical significance, e.g., p-value < 0.05).

| Dataset | Method | ResNet-50 | | | ViT-B/16 | | |
|---|---|---|---|---|---|---|---|
| | | Acc.↑ | CLIP↑ | Mpnet↑ | Acc.↑ | CLIP↑ | Mpnet↑ |
| Common-3k | CLIP-dissect | - | 0.7456 | 0.4161 | - | 0.7182 | 0.2718 |
| | **MICLIP (Ours)** | - | **0.7624*** | **0.4334*** | - | **0.7618*** | **0.4310*** |
| Common-10k | CLIP-dissect | - | 0.7656 | 0.4696 | - | 0.7342 | 0.3637 |
| | **MICLIP (Ours)** | - | **0.7885*** | **0.5029*** | - | **0.7786*** | **0.4748*** |
| Common-20k | CLIP-dissect | - | 0.7900 | 0.5257 | - | 0.7563 | 0.4376 |
| | **MICLIP (Ours)** | - | **0.8145*** | **0.5812*** | - | **0.8138*** | **0.5783*** |
| ImageNet-1k | Act-Values | 0.9940 | 0.9995 | 0.9983 | 0.9940 | 0.9989 | 0.9975 |
| | CLIP-dissect | 0.9560 | 0.9902 | 0.9746 | 0.9500 | 0.9881 | 0.9631 |
| | **MICLIP (Ours)** | **1.0000*** | **1.0000*** | **1.0000*** | **1.0000*** | **1.0000*** | **1.0000*** |

**Finding 1: MICLIP gives more precise interpretations given specific representation units.** In Table 1, our MICLIP can achieve the highest Accuracy, CLIP score and Mpnet Score among baselines, illustrating our precise localization of identified features. Furthermore, MICLIP outperforms CLIP-Dissect even when evaluated on concept set rather than ImageNet-1k, which MICLIP is trained on. This also suggests that MICLIP generalizes to broader scope of concepts.

Table 2: Accuracy deviations of enhancement and removal interventions on neurons and features. Best performing methods are highlighted in **bold**. Values that contradict the expected outcome (*e.g.*, enhancement leading to a decrease in accuracy) are marked in red.

(a) Intervention on neurons

| Method | Enhancement $\Delta Acc$ (%) ($\uparrow$) | | | Removal $\Delta Acc$ (%) ($\downarrow$) | | |
|---|---|---|---|---|---|---|
| | ResNet-50 | ViT-B/16 | CLIP | ResNet-50 | ViT-B/16 | CLIP |
| Act-Values | 2.27 ($\pm$ 0.03) | -0.19 ($\pm$ 0.01) | -8.05 ($\pm$ 0.02) | -8.98 ($\pm$ 0.08) | **-1.43 ($\pm$ 0.01)** | **-23.30 ($\pm$ 0.00)** |
| Network Dissection | 0.78 ($\pm$ 0.02) | **0.35 ($\pm$ 0.01)** | 0.23 ($\pm$ 0.02) | -2.95 ($\pm$ 0.15) | -0.37 ($\pm$ 0.03) | -0.88 ($\pm$ 0.05) |
| CLIP-dissect | 3.05 ($\pm$ 0.18) | 0.19 ($\pm$ 0.03) | -0.04 ($\pm$ 0.03) | -12.31 ($\pm$ 0.67) | -0.04 ($\pm$ 0.02) | -1.16 ($\pm$ 0.14) |
| V-Interp | 1.71 ($\pm$ 0.22) | -0.04 ($\pm$ 0.02) | -0.29 ($\pm$ 0.10) | -8.04 ($\pm$ 0.71) | -0.04 ($\pm$ 0.00) | -0.14 ($\pm$ 0.05) |
| **MICLIP (Ours)** | **5.32 ($\pm$ 0.03)** | 0.18 ($\pm$ 0.01) | **1.10 ($\pm$ 0.05)** | **-17.24 ($\pm$ 0.05)** | -0.04 ($\pm$ 0.02) | -1.50 ($\pm$ 0.08) |

(b) Intervention on SAE features

| Method | Enhancement $\Delta Acc$ (%) ($\uparrow$) | | | Removal $\Delta Acc$ (%) ($\downarrow$) | | |
|---|---|---|---|---|---|---|
| | ResNet-50 | ViT-B/16 | CLIP | ResNet-50 | ViT-B/16 | CLIP |
| Act-Values | **4.34 ($\pm$ 0.00)** | 3.68 ($\pm$ 0.02) | 0.43 ($\pm$ 0.08) | **-11.98 ($\pm$ 0.00)** | -22.77 ($\pm$ 0.11) | -15.94 ($\pm$ 0.06) |
| Network Dissection | 0.02 ($\pm$ 0.04) | 1.12 ($\pm$ 0.05) | 0.50 ($\pm$ 0.06) | -0.08 ($\pm$ 0.05) | -4.03 ($\pm$ 0.13) | -1.99 ($\pm$ 0.05) |
| CLIP-dissect | 2.27 ($\pm$ 0.09) | 5.04 ($\pm$ 0.05) | 4.85 ($\pm$ 0.03) | -7.30 ($\pm$ 0.03) | -27.78 ($\pm$ 0.09) | -11.05 ($\pm$ 0.12) |
| V-Interp | 0.91 ($\pm$ 0.02) | 1.90 ($\pm$ 0.01) | 1.33 ($\pm$ 0.00) | -2.88 ($\pm$ 0.09) | -7.55 ($\pm$ 0.06) | -2.83 ($\pm$ 0.00) |
| **MICLIP (Ours)** | 3.89 ($\pm$ 0.03) | **5.57 ($\pm$ 0.02)** | **5.88 ($\pm$ 0.03)** | -10.99 ($\pm$ 0.02) | **-32.04 ($\pm$ 0.20)** | **-17.70 ($\pm$ 0.02)** |

## 4.3 INTERVENTION FOR MODEL STEERING

**Analysis 2: Verifying Mechanism Localization via Intervention on Discriminative Models.** Effective mechanistic interpretability should enable meaningful interventions, either enhancing or suppressing the influence of concept-related mechanisms, thereby improving or degrading classification performance on the target concept. We verify our localization by intervening on the top-5 neurons or features (*i.e.*, $\tau = 5$ in Equation 3) for each ImageNet-1k concept. We measure the change in classification accuracy $\Delta Acc$ after applying either *enhancement* ($\times 2$ scaling) and *removal* ($\times 0$ scaling) after interventions on the activations of the following layers: the **10th**-layer for ViT-B/16 and CLIP/ViT-B-16, and the **stages.3.layers.1.shortcut** layer for ResNet-50.

**Finding 2: Our method, MICLIP, enables precise localization of the mechanisms that govern model classification.** For reference, the original classification accuracies of the models are 80.14% for ResNet-50, 80.32% for ViT-B/16, and 61.12% for CLIP. As detailed in Table 2, our MICLIP consistently enables a predictable and stable deviation in classification accuracy. In contrast, baselines like Act-Values exhibit an inconsistent response. Although they may show strong performance degradation upon *removal*, the same set of neurons often fails to enhance the model's performance. The ability of MICLIP to both enhance and suppress model performance using the same set of localized representation units provides strong evidence that we have successfully identified the true, functionally relevant neurons and features.

**Analysis 3: Verifying the Generalization of Mechanism Localization to Unseen Concepts via Intervention.** To evaluate generalization to unseen concepts, we repeat the intervention experiment in **Analysis 2** on the DTD texture

Table 3: Accuracy deviations from enhancement and removal interventions on neurons and features for unseen concepts. The best-performing methods are highlighted in **bold**.

(a) Intervention on neurons

| Method | Enhancement $\Delta Acc$ (%) ($\uparrow$) | Removal $\Delta Acc$ (%) ($\downarrow$) |
|---|---|---|
| CLIP-dissect | 0.00 ($\pm$ 0.15) | -0.65 ($\pm$ 0.28) |
| V-Interp | -0.39 ($\pm$ 0.24) | 0.06 ($\pm$ 0.37) |
| **MICLIP (Ours)** | **0.38 ($\pm$ 0.11)** | **-0.91 ($\pm$ 0.20)** |

(b) Intervention on SAE features

| Method | Enhancement $\Delta Acc$ (%) ($\uparrow$) | Removal $\Delta Acc$ (%) ($\downarrow$) |
|---|---|---|
| CLIP-dissect | 1.84 ($\pm$ 0.15) | **-5.04 ($\pm$ 0.08)** |
| V-Interp | 0.03 ($\pm$ 0.01) | -0.04 ($\pm$ 0.04) |
| **MICLIP (Ours)** | **2.00 ($\pm$ 0.10)** | -4.98 ($\pm$ 0.27) |

dataset, measuring the impact on CLIP's zero-shot classification accuracy. The original zero-shot classification accuracy of the CLIP model on this dataset is 44.80%.

**Finding 3: The mechanisms identified by MICLIP are semantically grounded and generalizable to unseen concepts.** As shown in Table 3, our MICLIP enables effective interventions on the CLIP/ViT-B-16 model, consistently enhancing or suppressing its zero-shot classification accuracy for unseen concepts. This is demonstrated by the predictable and stable changes observed in the model's performance metrics when we apply interventions to the localized neurons or features. Although MICLIP was trained solely on the ImageNet-1k dataset, its effectiveness in a zero-shot

setting highlights its strong generalization capabilities. These results confirm that our approach successfully localizes the key representation units governing model behavior, even for concepts not present in its original training data.

## 4.4 SEMANTIC ANALYSIS ON MICLIP

**Analysis 4: Unit Semantic Geometry in the Aligned Embedding Space.** In this part, we analyze the semantic geometry of representation units localized by MICLIP. This analysis aims to determine whether the localized SAE features exhibit semantic coherence as concepts. Similarly, we select concepts $c$ in ImageNet-1k, and then we use WordNet (Miller, 1995) to categorize these concepts into four categories ("mammal", "non-mammal", "tool" and "vehicle"). Following Huben et al. (2024), we extract localized features from $\mathbf{W}_{\text{dec}}$, *i.e.*, the learned dictionary. Then, we visualize 2D t-SNE (van der Maaten & Hinton, 2008) of the embeddings of these features in Figure 3. We provide more details in Appendix D.2.

**Finding 4: MICLIP learns a semantically coherent feature space.** Figure 3 shows that SAE features associated with related concepts form coherent clusters in MICLIP's aligned embedding space.

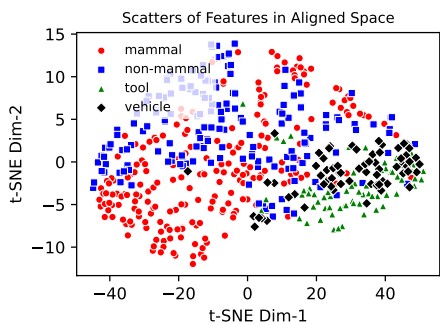

**Analysis 5: Visualization Verification of Localized Features via Attention Maps.** In this analysis, we investigate the spatial grounding of the learned features by examining their activations within the model's attention maps. We adopt the visualization method from DINO (Caron et al., 2021) to explore whether the SAE features we have localized correspond to a specific visual concept. Specifically, we leverage the self-attention map from **8th** layer of the CLIP/ViT-B-16 model, between the **CLS** token and all other image patch tokens. By preserving the out-

Figure 3: Features belonging to the same semantic category demonstrate compact clustering in the 2D embedding space.

put of specific localized features of image tokens, we generate a saliency map by computing the attention weights between the **CLS** token and image tokens. This map precisely highlights the regions of the input image that the features are attending to, providing a qualitative verification of the feature's intended visual semantics. The attention map shown in Figure 4 visualizes the spatial grounding of top-5 localized features for the first class **"kit fox"** in the ImageNet-1k metadata class order. The visualizations are specifically for the first six images of **"kit fox"** from the ImageNet-1k validation set. Additional results for other classes and images can be found in the Appendix E.1.

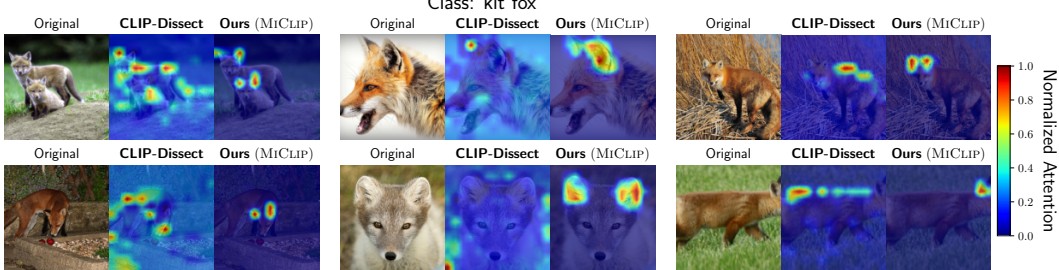

Figure 4: Visualizing the spatial grounding of top-5 localized SAE features of class **"kit fox"**. The attention map highlights the precise location of **"kit fox"** in ImageNet-1k (Russakovsky et al., 2015), confirming the feature's effective localization.

**Finding 5: MICLIP effectively localizes features that semantically correspond to specific visual concepts.** The attention map in Figure 4 highlights that our identified feature consistently activates around the ears of the **"kit fox"**, a key visual identifier for this class.

Table 4: Ablation Study: Accuracy deviations of enhancement and removal interventions on seen and unseen concepts. Best performing methods are in **bold**, second best underlined, contradictory values (*e.g.*, enhancement leading to an accuracy decrease) in red.

(a) Interventions on ImageNet-1k

| Method | Enhancement ΔAcc (%) (↑) | | | Removal ΔAcc (%) (↓) | | |
|---|---|---|---|---|---|---|
| | ResNet-50 | ViT-B/16 | CLIP | ResNet-50 | ViT-B/16 | CLIP |
| *Neurons* | | | | | | |
| Input-Only | 3.71 (± 0.02) | 0.12 (± 0.07) | 0.31 (± 1.02) | -12.63 (± 0.11) | -0.18 (± 0.06) | **-4.09 (± 2.51)** |
| Output-Only | **5.71 (± 0.01)** | **0.21 (± 0.03)** | 0.44 (± 0.41) | **-17.98 (± 0.04)** | **-0.26 (± 0.01)** | -1.17 (± 0.03) |
| MiCLIP | 5.32 (± 0.03) | 0.18 (± 0.01) | **1.10 (± 0.05)** | -17.24 (± 0.05) | -0.04 (± 0.02) | -1.50 (± 0.08) |
| *SAE Features* | | | | | | |
| Input-Only | 2.52 (± 0.05) | 4.47 (± 0.88) | 5.49 (± 0.04) | -8.11 (± 0.05) | -24.29 (± 5.97) | -15.25 (± 0.14) |
| Output-Only | **4.05 (± 0.01)** | **5.69 (± 0.03)** | 5.32 (± 0.03) | **-11.17 (± 0.01)** | **-32.79 (± 0.04)** | -16.59 (± 0.08) |
| MiCLIP | 3.89 (± 0.03) | 5.57 (± 0.02) | **5.88 (± 0.03)** | -10.99 (± 0.02) | -32.04 (± 0.20) | **-17.70 (± 0.02)** |

(b) Interventions on DTD

| Method | Enhancement ΔAcc (%) (↑) | Removal ΔAcc (%) (↓) |
|---|---|---|
| *Neurons* | | |
| Input-Only | **0.69 (± 0.36)** | **-5.18 (± 1.62)** |
| Output-Only | -2.60 (± 2.50) | -0.53 (± 0.30) |
| MiCLIP | 0.38 (± 0.11) | -0.91 (± 0.20) |
| *SAE Features* | | |
| Input-Only | **2.84 (± 0.16)** | **-6.85 (± 0.16)** |
| Output-Only | 1.15 (± 0.01) | -2.96 (± 0.03) |
| MiCLIP | 2.00 (± 0.10) | -4.98 (± 0.27) |

## 4.5 ABLATION STUDY

**Analysis 6: Ablation Study on Input- and Output-Grounded Alignment.** To dissect the contributions of our two alignment components, we perform an ablation study with three variants: MiCLIP (Input-Only), MiCLIP (Output-Only), and the full MiCLIP. We replicate the intervention experiments on seen concepts from ImageNet-1k (Analysis 2) and unseen texture concepts from DTD (Analysis 3), evaluating effectiveness by the accuracy change $\Delta Acc$ after *enhancement* ($\times 2$) and *removal* ($\times 0$) of the top-5 localized mechanisms.

**Finding 6: The combination of input- and output-grounded alignment is crucial for robust and generalizable mechanistic interpretability.** The results, presented in Table 4, reveal a clear synergy between the two alignment strategies. The Output-Only performs strongly on seen ImageNet-1k concepts but fails to generalize to the unseen DTD dataset, as evidenced by its poor performance and contradictory results (*e.g.*, enhancement causing an accuracy drop). Conversely, the Input-Only shows better generalization to DTD but is less effective on ImageNet-1k. The full MiCLIP achieves strong performance on seen concepts while maintaining robust generalization to novel ones, confirming that integrating both alignment strategies is essential.

## 4.6 UNDERSTANDING FLAWED VISUAL REASONING IN MODEL PREDICTIONS

We leverage MiCLIP to diagnose failures in visual reasoning in the CLIP/ViT-B-16 model by tracing the semantic trajectory of an image's internal representations across layers. This is achieved by projecting layer-specific activations into a semantically aligned embedding space via the neuron encoder $E_n$. We then compute cosine similarities to text embeddings of the **ground-truth (GT)** and **misclassified labels in the incorrect case** using the CLIP text encoder $E_c$. For the incorrect case, the predicted label is the misclassification, which is also used for comparison in the correct case. As illustrated in Figure 5, presenting an example with a GT of "sea anemone" and a misclassification of "feather boa". Correct predictions maintain a higher GT similarity across all residual layers. Erroneous predictions show a clear point where predicted-label similarity surpasses GT, pointing out where the model's view shifts. With this diagnose method, MICLIP is able to trace the semantic trajectory of the reasoning process, and localize the layer where the representation shifts from the correct concept to the incorrect one.

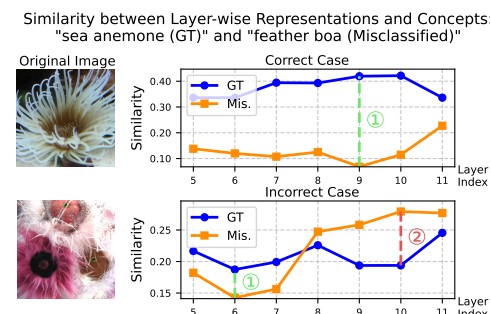

Figure 5: The plot shows cosine similarities between layer-wise representation embeddings and text embeddings of the ground-truth (GT; blue) and misclassified label (Mis.; orange). ① marks the layer where GT dominates most, while ② marks where Mis. overtakes GT, revealing the failure point.

## 4.7 VISUALIZING LOW-LEVEL FEATURE PRIMITIVES

**Analysis 7: Visualization Verification of Low-Level Features.** In this analysis, we extend our

investigation to ascertain whether our learned features also correspond to low-level visual primitives, such as colors, shapes and textures. To explore the spatial grounding of these elementary features, we employ the same attention map visualization technique detailed in Analysis 5, specifically utilizing the attention maps from relatively low-level layers (i.e., the *3rd*) layer of the CLIP/ViT-B-16 model.

While the ImageNet-1k dataset lacks explicit labels for such primitives, we identified representative samples by selecting images that exhibited high activations for certain localized features discovered by MICLIP. For instance, we found features that systematically activate on images containing prominent green regions, and indeed, their top-activating images from the dataset consistently share this specific visual characteristic. The attention maps shown in Figure 6 visualize the spatial grounding for this feature, confirming its specialization for the color **"green"**. This provides a qualitative verification that our method can effectively identify and localize not only high-level semantic concepts but also fundamental visual building blocks. Additional results for other low-level features, such as a shape-detecting feature for grids, can be found in Appendix E.3.

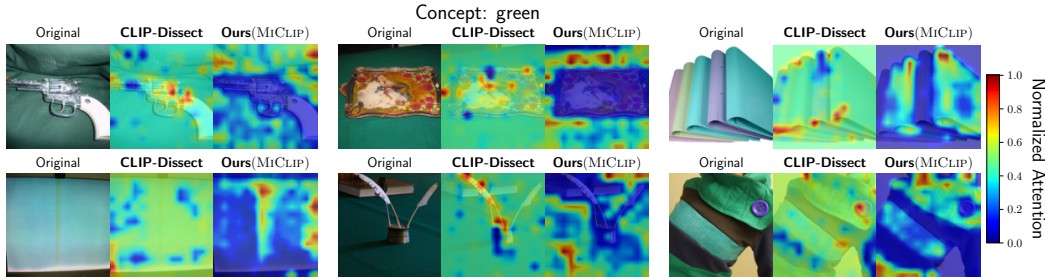

Figure 6: Visualizing the spatial grounding of a localized SAE feature for the color **"green"**. The attention maps highlight that this feature consistently activates on green regions across different images from ImageNet-1k, confirming its effective localization of this low-level visual property.

**Finding 7: MICLIP effectively localizes low-level features that correspond to fundamental visual primitives.** As demonstrated in Figure 6, the attention maps for the identified "green" feature consistently and precisely highlight green-colored regions, regardless of the object's semantics. This confirms that our method can ground foundational visual concepts, like specific colors, to their corresponding spatial locations within an image.

## 5    CONCLUSION

We propose MICLIP, a representation-based automated mechanistic interpretability framework that bridges the gap between the internal mechanisms of vision models and human-understandable concepts, through aligning them in a shared semantic embedding space. MICLIP eliminates the reliance on the activation-magnitude assumption and jointly leverages semantic signals from both the inputs and outputs of vision models, thereby capturing a more comprehensive and faithful view of the model's reasoning process. By leveraging contrastive learning to align internal features with semantic concepts, MICLIP provides fine-grained and precise mechanistic interpretation across diverse vision models. Coupled with external modules like k-SAE, it establishes a universal framework supporting both neuron- and feature-level analysis. Extensive qualitative and quantitative experiments confirm its effectiveness in interpreting, localizing, and steering internal features, while also demonstrating generalization to unseen concepts, applicability across diverse architectures, and utility in analyzing flawed model behaviors.

Despite these advancements, we acknowledge certain limitations. First, MICLIP's interpretability fidelity is inherently bounded by the semantic coverage and potential biases of the pre-trained CLIP embedding space. However, our empirical observations indicate that CLIP's classification biases do not necessarily hamper MICLIP's intervention capabilities; valid features can still be identified and steered when CLIP performs poorly. Integrating orthogonal advancements to mitigate these biases could further enhance fidelity. Second, current validation is restricted to discriminative vision models. Future work could extend this approach to generative models, broadening the applicability and impact of mechanistic interpretability in real-world AI systems.

## 6 ACKNOWLEDGEMENT

This research received support from National Natural Science Foundation of China (Grant No. 62406193). This work was also supported by ShanghaiTech AI Initiative (Grant No. AI2026B08). The authors gratefully acknowledge further assistance provided by Shanghai Frontiers Science Center of Human-centered Artificial Intelligence, MoE Key Lab of Intelligent Perception and Human-Machine Collaboration, and HPC Platform of ShanghaiTech University.

## 7 ETHICS STATEMENT

This research complies with the ICLR ethical guidelines, upholding transparency, reproducibility, and responsible use of AI. Our work aims to make vision models more transparent and interpretable, contributing to society by enabling safer and more responsible AI systems. We believe that increasing model transparency benefits a broad range of stakeholders and helps mitigate potential misuse. This research does not involve human subjects, personal data, or sensitive information, and thus poses no risks to privacy, health, or safety.

We have faithfully reported our findings without fabrication or falsification. All datasets, baselines, and related works are properly cited, and our methods are designed to be transparent and reproducible. We have discussed the limitations of our work and possible future extensions.

We appreciate prior contributions in mechanistic interpretability and aim for our work to further promote fairness, accountability, and responsible AI deployment.

## 8 REPRODUCIBILITY STATEMENT

To ensure the reproducibility of our work, this paper provides comprehensive details regarding our experimental setup, models, and evaluation protocols. Specifically:

- **Models and Datasets:** The target models (ResNet-50, ViT-B-16, and CLIP/ViT-B-16) and datasets (ImageNet-1k and DTD) used for our experiments are explicitly stated in Section 4.1.
- **Implementation Details:** Key details for implementing our method, MICLIP, are provided in Section 4.1. This includes the training data size (100,000 images from ImageNet-1k), the use of frozen CLIP encoders, the architecture of the neuron encoder ($E_n$), and the choice of k-SAE for feature disentangling. Further details on baseline methods are available in Appendix D.1.
- **Experimental Procedures:** Each analysis is described with its specific protocol. For instance, the quantitative evaluation framework is detailed in Section 4.2 (Analysis 1). The intervention strategy, including the specific layers targeted and scaling factors, is described in Section 4.3 (Analysis 2). All other analyses are similarly detailed in their respective sections.

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

## A USAGE OF LARGE LANGUAGE MODELS (LLMS)

In this work, LLMs are used in some automated interpretability baselines (Zhang et al., 2024) to generate or summarize textual explanations. Our proposed method does not rely on LLMs, and their use is restricted to fair comparison with prior approaches.

During paper writing, LLMs are only used to polish the language and use of words. We promise they are **not** used to generate ideas, methodology design, and experimental analysis.

## B MICLIP ALGORITHMS

In this section, we elaborate on the methodology of our MICLIP, as introduced in Section 3.1 and Section 3.2. For clarity, we present two algorithms that outline the key steps of the approach in detail.

### B.1 CONTRASTIVE LEARNING FOR ALIGNMENT

We introduce the detailed steps of our contrastive learning (Section 3.1) in Algorithm 1. We only train the neuron encoder $E_n(\cdot; \theta_n)$.

---

**Algorithm 1** Mechanism-Concept Alignment using CLIP-based Encoders

---

**Input:** Input image set $\mathcal{X} = \{x_i\}_{i=1}^N$, activation set $\mathcal{A} = \{\mathbf{a}_i\}_{i=1}^N$, predicted output set $\{\hat{c}_i\}_{i=1}^N$
Pretrained CLIP visual encoder $E_i$ and text encoder $E_c(\cdot)$.
Batch size $B$, step size $\eta$.
**Output:** Trained parameter $\theta_n$ of neuron encoder $E_n(\cdot; \theta_n)$.

1: $\theta_n^0 \leftarrow \text{RandomInit}(\theta_n)$  $\qquad \triangleright$ *Initialization.*
2: **for** $t = 1, \cdots, T$ **do**  $\qquad \triangleright$ *Batch training, terminate until $\theta_n$ converges.*
3: $\quad \{(x_i, \mathbf{a}_i, \hat{c}_i)\}_{i=1}^B \leftarrow \text{SampleBatch}(\mathcal{D}, B)$

4: $\quad$ **for** $i = 1, \cdots, B$ **do**  $\qquad \triangleright$ *Compute embeddings.*
5: $\quad\quad \mathbf{n}_i \leftarrow E_n(\mathbf{a}_i; \theta_n^{t-1})$  $\qquad \triangleright$ *Section 3.1*
6: $\quad\quad \mathbf{c}_i \leftarrow E_c(\hat{c}_i)$
7: $\quad\quad \mathbf{x}_i \leftarrow E_i(x_i)$
8: $\quad$ **for** $(i, j) \in \{1, \cdots, B\}^2$ **do**  $\qquad \triangleright$ *Compute scores.*
9: $\quad\quad S_{ij}^c \leftarrow \frac{\mathbf{n}_i \cdot \mathbf{c}_j}{\|\mathbf{n}_i\| \cdot \|\mathbf{c}_j\|}$
10: $\quad\quad S_{ij}^x \leftarrow \frac{\mathbf{n}_i \cdot \mathbf{x}_j}{\|\mathbf{n}_i\| \cdot \|\mathbf{x}_j\|}$

11: $\quad \mathcal{L}_{\text{CLIP}}^{\text{out}} \leftarrow -\frac{1}{2B} \sum_{i=1}^B \log \frac{e^{S_{ii}^c}}{\sum_{j=1}^B e^{S_{ij}^c}} - \frac{1}{2B} \sum_{j=1}^B \log \frac{e^{S_{jj}^c}}{\sum_{i=1}^B e^{S_{ij}^c}}$  $\quad \triangleright$ *Neuron-concept loss*

12: $\quad \mathcal{L}_{\text{CLIP}}^{\text{in}} \leftarrow -\frac{1}{2B} \sum_{i=1}^B \log \frac{e^{S_{ii}^x}}{\sum_{j=1}^B e^{S_{ij}^x}} - \frac{1}{2B} \sum_{j=1}^B \log \frac{e^{S_{jj}^x}}{\sum_{i=1}^B e^{S_{ij}^x}}$  $\quad \triangleright$ *Neuron-image loss*

13: $\quad \mathcal{L}_{\text{alignment}} \leftarrow \mathcal{L}_{\text{CLIP}}^{\text{out}} + \mathcal{L}_{\text{CLIP}}^{\text{in}}$  $\qquad \triangleright$ *Equation 2*

14: $\quad \theta_n^t \leftarrow \theta_n^{t-1} + \eta \nabla_{\theta_n} \mathcal{L}_{\text{alignment}}$  $\qquad \triangleright$ *Update. GD as an example.*
15: $\theta_n \leftarrow \theta_n^T$
16: **Return:** $\theta_n$.

---

### B.2 LOCALIZATION AND DESCRIPTION OF REPRESENTATION UNITS

We also describe the detailed steps of our feature localization and descriptions (Section 3.2) in Algorithm 2. For interpretation, we can get $\mathbf{L}_c$ (top $\tau$ feature indices for concept $c$) or $\mathbf{D}_{u_i}$ (top $\tau$ concepts for unit $u_i$).

---

**Algorithm 2** Representation-Based Mechanism-Concept Identification

---

**Input:** Feature vector $\mathbf{f} \in \mathbb{R}^m$ for input $\mathbf{x}$ (from k-SAE, optional).

Concept set $\mathcal{C}$.

Dictionary $\mathbf{W}_{\text{dec}} \in \mathbb{R}^{n \times m}$, bias $\mathbf{b}_{\text{pre}} \in \mathbb{R}^n$.

Pre-trained encoders $\text{E}_n$, $\text{E}_c$ .

Hyper-parameter $\tau$.

One-hot vectors $e^{(i)} = [0, \cdots, \underbrace{1}_{i\text{-th position}}, \cdots, 0]^\top \in \mathbb{R}^m, \quad i = 1, \cdots, m.$

(Optional) Specific concept $c \in \mathcal{C}$ (for localization) or feature index $i$ (for description).

(Optional) Feature vector $\mathbf{f} \in \mathbb{R}^m$ for input $\mathbf{x}$ (from k-SAE).

**Output:** $\mathbf{L}_c$ (top $\tau$ unit indices for concept $c$) or $\mathbf{D}_{f_i}$ (top $\tau$ concepts for unit $u_i$).

1:  **function** CONCEPT-TO-MECHANISM LOCALIZATION($c$)                 ▷ *Given concept c.*
2:      $\mathbf{c} \leftarrow \text{E}_c(c)$
3:      **for** $i = 1$ to $m$ **do**
4:          **if** $\mathbf{f} \neq$ **None then**                      ▷ *Localize SAE feature*
5:              $\bar{\mathbf{a}}_i \leftarrow \mathbf{W}_{\text{dec}}(\mathbf{f} \odot e^{(i)}) + \mathbf{b}_{\text{pre}}$       ▷ *Equation 10*
6:          **else**                                                    ▷ *Localize neuron*
7:              $\bar{\mathbf{a}}_i \leftarrow e^{(i)}$
8:          $\bar{\mathbf{n}}_i \leftarrow \text{E}_n(\bar{\mathbf{a}}_i)$
9:          $\textbf{sim}(u_i, c) \leftarrow \frac{\bar{\mathbf{n}}_i \cdot \mathbf{c}}{\|\bar{\mathbf{n}}_i\| \|\mathbf{c}\|}$       ▷ *Compute relevance score.*
10:     $\mathbf{L}_c \leftarrow \underset{i}{\text{SelectTop-}\tau} \{\textbf{sim}(u_i, c)\}_{i=1}$       ▷ *Equation 3*
11:     **return** $\mathbf{L}_c$

12: **function** MECHANISM-TO-CONCEPT DESCRIPTION($i$)                 ▷ *Given feature index i.*
13:     **if** $\mathbf{f} \neq$ **None then**                         ▷ *Describe SAE feature*
14:         $\bar{\mathbf{a}}_i \leftarrow \mathbf{W}_{\text{dec}}(\mathbf{f} \odot e^{(i)}) + \mathbf{b}_{\text{pre}}$       ▷ *Equation 10*
15:     **else**                                                       ▷ *Describe neuron*
16:         $\bar{\mathbf{a}}_i \leftarrow e^{(i)}$
17:     $\bar{\mathbf{n}}_i \leftarrow \text{E}_n(\bar{\mathbf{a}}_i)$
18:     **for** $c_j \in \mathcal{C}$ **do**
19:         $\mathbf{c}_j \leftarrow \text{E}_c(c_j)$
20:         $\textbf{sim}(u_i, c_j) \leftarrow \frac{\bar{\mathbf{n}}_i \cdot \mathbf{c}_j}{\|\bar{\mathbf{n}}_i\| \|\mathbf{c}_j\|}$       ▷ *Compute relevance score.*
21:     $\mathbf{D}_{u_i} \leftarrow \underset{j}{\text{SelectTop-}\tau} \{\textbf{sim}(u_i, c_j)\}_{c_j \in \mathcal{C}}$       ▷ *Equation 4*
22:     **return** $\mathbf{D}_{u_i}$

---

## B.3 THEORETICAL CONSISTENCY OF SINGLE-UNIT LOCALIZATION.

A potential concern regarding the localization phase is whether applying the neuron encoder $\text{E}_n$ to a single basis vector $e^{(i)}$ (representing a single neuron or feature) constitutes an Out-of-Distribution (OOD) shift, given that the encoder is trained on dense activation vectors $\mathbf{a}$. We demonstrate here that the linearity of the projection $\text{E}_n$ guarantees that the training and testing phases are mathematically consistent.

Recall that the training objective is based on the symmetric InfoNCE loss. Focusing on the neuron-image alignment term, for a positive pair $(\mathbf{n}, \mathbf{x})$, where $\mathbf{n} = \text{E}_n(\mathbf{a})$, the loss in one direction is:

$$\ell_{\text{InfoNCE}} = -\log \frac{\exp(\text{sim}(\mathbf{n}, \mathbf{x})/\tau)}{\sum_{\mathbf{x}'} \exp(\text{sim}(\mathbf{n}, \mathbf{x}')/\tau)}. \tag{6}$$

The key to resolving the OOD concern lies in the linearity of $\text{E}_n$. The activation vector is $\mathbf{a} = \sum_{i=1}^n a_i \cdot e^{(i)} \in \mathbb{R}^n$, where $\{e^{(i)}\}$ is the standard basis. Due to linearity, the output $\mathbf{n}$ decomposes into projected basis vectors:

$$\mathbf{n} = \text{E}_n(\mathbf{a}) = \text{E}_n\left(\sum_i a_i e^{(i)}\right) = \sum_i a_i \underbrace{\text{E}_n(e^{(i)})}_{\text{Single Unit Embedding}}. \tag{7}$$

To analyze the similarity composition, following prior work on semantic decomposition (**?**), we assume the CLIP image embedding space is spanned by a dictionary of semantic basis vectors $\{v_k\}$, such that an image is represented as $\mathbf{x} = \sum_k k_k \cdot v_k$, where $k_k$ are the coefficients. Substituting these decompositions into the alignment calculation, the similarity function becomes a weighted sum of component-wise similarities:

$$\mathrm{sim}(\mathbf{n}, \mathbf{x}) \propto \mathrm{sim}\left(\sum_i a_i \mathrm{E_n}(e^{(i)}), \sum_k k_k v_k\right) = \sum_{i,k} a_i k_k \cdot \underbrace{\mathrm{sim}(\mathrm{E_n}(e^{(i)}), v_k)}_{\text{Single Neuron-Semantic Align}} . \tag{8}$$

This decomposition demonstrates that even though the training inputs $\mathbf{a}$ are combinations of multiple features, the contrastive loss explicitly maximizes the correlation between the specific activation component (the projected basis vector $\mathrm{E_n}(e^{(i)})$) and the specific visual feature $v_k$ present in the image. Therefore, the single-neuron projection $\mathrm{E_n}(e^{(i)})$ is explicitly optimized during the contrastive training process, ensuring it is not OOD during analysis.

**Scale Invariance of Mechanism Analysis.** A related consideration is whether investigating vectors of unit length is valid, given that the scaling of neuron activations is not standardized. We clarify that our method is robust to this because we employ **cosine similarity** for all alignment measurements in both training and inference stages:

$$\mathrm{sim}(\mathbf{u}, \mathbf{c}) = \frac{\langle \mathbf{u}, \mathbf{c} \rangle}{\|\mathbf{u}\| \cdot \|\mathbf{c}\|}. \tag{9}$$

Cosine similarity is inherently independent of vector magnitude. Since $\mathrm{E_n}$ is a linear projection, scaling the input vector by a scalar $\alpha$ results in a linearly scaled embedding: $\mathrm{E_n}(\alpha e^{(i)}) = \alpha \mathrm{E_n}(e^{(i)})$. When calculating similarity, this scalar $\alpha$ appears in both the numerator (dot product) and the denominator (norm) and cancels out. Thus, our analysis relies solely on the angular direction of the embeddings, making the method invariant to the magnitude or scaling of the original neuron activations.

## C  IMPLEMENTATION DETAILS

### C.1  FEATURE DISENTANGLEMENT

In addition to aligning model neurons with semantic concepts, MICLIP can adopt widgets to disentangle the neuron space into a sparse feature space, to address the issue of *polysemanticity* Mu & Andreas (2020), where a single neuron may encode multiple unrelated concepts. In this section, we introduce k-Sparse Autoencoders (k-SAE) Makhzani & Frey (2014) that transform obscure neuron activations into sparse features that expose more interpretable relationships with human-understandable semantic concepts. It is applicable together with MICLIP, as mentioned in Section 3.2.

For a representation $\mathbf{a} \in \mathbb{R}^n$ from one specific layer in the model, k-SAE follows the dictionary learning paradigm and conducts encoding and decoding stages as shown as follows:

$$\mathbf{f} = [v_1, v_2, \ldots, v_m] = \underbrace{\text{Top-k}(\mathbf{W}_{\text{enc}}(\mathbf{a} - \mathbf{b}_{\text{pre}}))}_{\text{encoding}}, \quad \hat{\mathbf{a}} = \underbrace{\mathbf{W}_{\text{dec}}\mathbf{f} + \mathbf{b}_{\text{pre}}}_{\text{decoding}} . \tag{10}$$

The k-SAE encodes $\mathbf{a}$ into an $m$-dimensional feature $\mathbf{f} \in \mathbb{R}^m$ and then reconstructs it as $\hat{\mathbf{a}}$, a combination of the features, where the encoder and decoder matrix $\mathbf{W}_{\text{enc}} \in \mathbb{R}^{m \times n}, \mathbf{W}_{\text{dec}} \in \mathbb{R}^{n \times m}$ and the bias $\mathbf{b}_{\text{pre}} \in \mathbb{R}^n$ are learnable parameters, denoted together by $\theta_{\text{SAE}}$. Top-k$(\cdot)$ is an operator that selects the top k largest feature values and leaves them unchanged while setting the remaining features to zero. This introduces the sparsity constraint on the features Makhzani & Frey (2014), requiring $\|\mathbf{f}\|_0 \leq k$.

The k-SAE is trained by minimizing the representation reconstruction error while enforcing the feature sparsity constraint. The objective is formulated in Equation 11.

$$\underset{\theta_{\text{SAE}}}{\text{minimize}} \quad \mathcal{L}(\theta_{\text{SAE}}; \mathbf{a}) = \|\mathbf{a} - \hat{\mathbf{a}}\|_2^2 , \text{ subject to } \|\mathbf{f}\|_0 \leq k. \tag{11}$$

Table 5: Comparison of computational cost (FLOPs) and resource usage to analyze mechanisms over 100,000 images using ViT-B-16. "-" or "N/A" indicates no training phase is required.

| Method | FLOPs (T-FLOPs) | | | | Time | | Memory | |
| | Training | Neuron Loc. | Feature Loc. | Total | Training | Inference | VRAM | Storage |
|---|---|---|---|---|---|---|---|---|
| **MICLIP (Ours)** | 7290.39 | 0.10 | 186.60 | 7477.09 | 30 min | **2 min** | 16 GB | **94 MB** |
| Act-Values | - | 3524.01 | 4256.42 | 7780.43 | - | 30 min | 16 GB | 97 MB |
| CLIP-Dissect | - | 7270.72 | 7769.20 | 15039.92 | - | 30 min | 16 GB | 9.7 GB |

## C.2 TRAINING DETAILS

**Training Details of Mechanism-Concept Alignment.** Our MICLIP was trained for a single epoch on a subset of 100,000 images sampled from the ImageNet-1k training set. We employed the **Adam optimizer** with a fixed learning rate of $1 \times 10^{-4}$. The training was performed with a batch size of 1024.

Alignment Loss Curves for Different Target Models

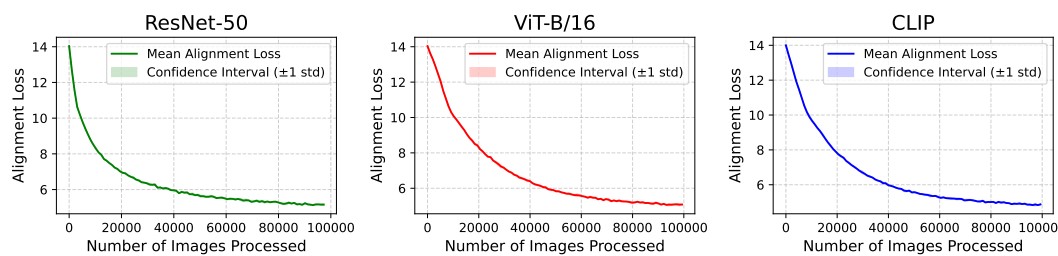

Figure 7: Alignment loss curves for three target models. The curves represent the mean alignment loss and its standard deviation calculated across three random seeds for each model during training.

**Training Details of k-SAE.** Our k-SAE was trained on the entire ImageNet-1k training set. We employed the Adam optimizer with a fixed learning rate of $1 \times 10^{-3}$. The training was performed with a batch size of 32 (see Figure 8).

**Training Efficiency.** As shown in Table 5, our MICLIP achieves comparable training and inference efficiency to the baselines, despite requiring a one-time training phase.

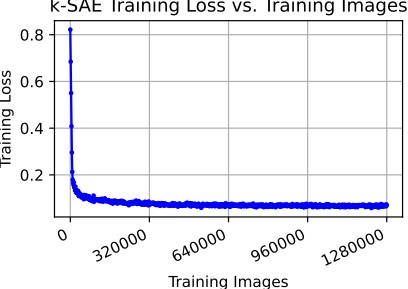

Figure 8: Training loss curve of k-SAE for ViT-B-16 model.

## D EXPERIMENTAL SETTINGS

### D.1 BASELINES

**Act-Values** is a fundamental baseline identifies class-specific mechanisms (neurons or features) by ranking them based on their average activation for a given class. The top-ranked mechanisms are then used for intervention tasks (Section 4.3).

**Network Dissection** (Bau et al., 2017) is a classic interpretability method that identifies concept detectors by measuring the alignment between a neuron's or feature's activation map and a concept's semantic segmentation. We follow its core principle of using Intersection over Union (IoU) as the alignment metric. However, to enable analysis on a broader range of concepts in ImageNet-1k (Russakovsky et al., 2015), we leverage Grounded SAM(Ren et al., 2024) to generate concept masks.

**CLIP-Dissect** (Oikarinen & Weng, 2023) uses CLIP to describe the function of individual neurons. For our description accuracy comparison (Section 4.2), we follow its original procedure. For intervention experiments (Section 4.3), we rank the neurons or features identified by CLIP-Dissect, then select those with the highest similarities for a target class.

**V-Interp** (Zhang et al., 2024) is inspired by text-based methods where large models explain smaller ones (Bills et al., 2023; Paulo et al., 2024). We implement a visual equivalent based on (Zhang et al., 2024). This approach identifies the image patches that cause the highest neuron or feature activations and feeds them to a powerful "explainer" Large Multimodal Model (LMM). The LMM's zero-shot descriptions of these patches serve as the final captions of neurons or features. Since this process yields a free-text description for each neuron or feature, we then use the CLIP/ViT-B-16 text encoder to select the unit whose description is most semantically similar to a given target concept.

## D.2 SEMANTIC ANALYSIS

### D.2.1 VISUALIZATION OF LOCALIZATION FEATURES

In this section, we provide details for the analysis of semantic geometry of SAE features in MI-CLIP's learned embedding space. We select a set of concepts from the labels of ImageNet-1k, then categorize them into four groups using WordNet (Miller, 1995) that are "mammal", "non-mammal'', "tool", "vehicle".

To be more specific, "mammal" and "vehicle" refer to synset "mammal.n.01" and "vehicle.n.01" respectively; "non-mammal" refers to the combination of "bird.n.01", "fish.n.01", "reptile.n.01", "amphibian.n.01" and "invertebrate.n.01". "tool" refers to the combination of synsets "tool.n.01", "appliance.n.01", "furniture.n.01" and "instrument.n.01".

### D.2.2 ATTENTION MAP VISUALIZATION

To qualitatively verify that our localized SAE features correspond to distinct and correct regions of an input image, we introduce a visualization technique based on feature-conditioned attention. This method adapts the attention map visualization from DINO (Caron et al., 2021) but introduces a critical intervention step. Instead of merely observing the model's natural attention, we constrain the model's activations to lie within the subspace of specific, pre-selected features. This allows us to generate a saliency map that directly reveals the spatial regions the model focuses on when its reasoning is guided only by the features associated with a given concept.

The process consists of three main steps: a partial forward pass, a targeted activation intervention, and the generation of the final attention map.

1. **Partial Forward Pass and Activation Extraction**
   Given an input image $x$ and a target intervention layer $l$, we perform a forward pass up to that layer. We extract the activations for all tokens, separating the activation vector for the **CLS** token, $\mathbf{a}_{\text{CLS}}^{(l)} \in \mathbb{R}^n$, from the matrix of activation vectors for the $P$ image patch tokens, $\mathcal{A}_{\text{patches}}^{(l)} \in \mathbb{R}^{P \times n}$. *Our intervention is applied exclusively to the patch tokens; the* **CLS** *token's activation remains unmodified.*

2. **Activation Intervention on Patch Tokens**
   This step is the core of our method. Given a set of $\tau$ SAE feature indices $\{i\}_{i \in \mathbf{L}_c}$ that have been localized to a concept $c$ (using Equation 3), we intervene on the patch activations by passing them through the SAE and filtering the resulting sparse code.

   First, each patch's activation vector $\mathbf{a}_p^{(l)}$ from $\mathcal{A}_{\text{patches}}^{(l)}$ is passed through the pre-trained SAE encoder, $\mathbf{W}_{\text{enc}}$, to obtain its corresponding sparse feature activation vector $\mathbf{z}_p \in \mathbb{R}^{d_s}$, where $d_s$ is the dictionary size:
   $$\mathbf{z}_p = \mathbf{W}_{\text{enc}}(\mathbf{a}_p^{(l)} - \mathbf{b}_{\text{pre}}) \tag{12}$$
   Next, we generate a masked sparse vector, $\tilde{\mathbf{z}}_p$, by retaining only the activation values at the indices corresponding to our target concept and setting all others to zero. This can be expressed as an element-wise product with a mask vector $\mathbf{m}_c \in \{0, 1\}^{d_s}$, where $(\mathbf{m}_c)_i = 1$ if $i \in \mathbf{L}_c$ and 0 otherwise:
   $$\tilde{\mathbf{z}}_p = \mathbf{z}_p \odot \mathbf{m}_c \tag{13}$$
   Finally, this filtered sparse code is passed through the SAE decoder, $\mathbf{W}_{\text{dec}}$, to reconstruct the activation vector in the original activation space. This yields the modified patch activation vector $\tilde{\mathbf{a}}_p^{(l)}$:
   $$\tilde{\mathbf{a}}_p^{(l)} = \mathbf{W}_{\text{dec}}\tilde{\mathbf{z}}_p + \mathbf{b}_{\text{pre}} \tag{14}$$

This operation is applied to all patch tokens, resulting in a matrix of modified activations, $\tilde{\mathcal{A}}_{\text{patches}}^{(l)} = [\tilde{\mathbf{a}}_1^{(l)}, \ldots, \tilde{\mathbf{a}}_P^{(l)}]^\top$.

3. **Feature-Conditioned Attention Map Generation**

   The full set of activations for subsequent layers is reassembled by concatenating the original, unmodified **CLS** activation with the modified patch activations. The forward pass then resumes from this reassembled state. We compute the self-attention weights in a subsequent block, focusing on the attention between the **CLS** token and the now-modified patch tokens.

   For a specific attention head $h$ (out of $H$ total heads), the attention weights $\boldsymbol{\alpha}^{(h)} \in \mathbb{R}^P$ are calculated as:

   $$\boldsymbol{\alpha}^{(h)} = \text{softmax}\left( \frac{(\mathbf{q}_{\text{CLS}}^{(h)})^\top \mathbf{K}_{\text{patches}}^{(h)}}{\sqrt{d_k}} \right) \tag{15}$$

   where the query vector $\mathbf{q}_{\text{CLS}}^{(h)} \in \mathbb{R}^{d_k}$ is derived from the original **CLS** activation $\mathbf{a}_{\text{CLS}}^{(l)}$, and the matrix of key vectors $\mathbf{K}_{\text{patches}}^{(h)} \in \mathbb{R}^{P \times d_k}$ is derived from the intervened patch activations $\tilde{\mathcal{A}}_{\text{patches}}^{(l)}$.

   The final 2D saliency map $\mathbf{S}$ is generated by averaging these attention weights across all attentions heads and reshaping the resulting $P$-dimensional vector to match the spatial layout of the image patches.

# E  MORE RESULTS

## E.1  SEMANTIC ANALYSIS

To further demonstrate the effectiveness and robustness of our method in localizing features that correspond to specific visual concepts, this appendix provides a more extensive set of examples across various classes. As detailed in the **Analysis 5**, we verify the spatial grounding of our learned features by visualizing their corresponding attention maps.

We follow the same visualization methodology described in **Analysis 5**. Specifically, we leverage the self-attention maps from the **8th** layer of the CLIP/ViT-B-16 model, computed between the **CLS** token and all image patch tokens. The resulting saliency maps highlight the precise image regions to which the identified features attend. The visualizations shown in Figure 9 and Figure 10 are generated for top-localized features across a diverse set of classes, using images from the ImageNet-1k validation set.

The results in Figure 9 and Figure 10 demonstrate that our MICLIP consistently localizes features to the relevant semantic regions across a wide array of classes. For each class, the attention maps reliably highlight the areas that define the core visual concept, and this localization remains stable across different images within that class. This consistency underscores the robustness of our approach, confirming that the identified features are not artifacts of specific images but are genuinely tied to the underlying visual semantics of the category.

## E.2  ADDITIONAL CASE STUDIES OF UNDERSTANDING FLAWED VISUAL REASONING

In this section, we provide additional case studies to further validate the diagnostic capabilities of MICLIP, as introduced in Section 4.6. Our approach pinpoints reasoning failures within the CLIP/ViT-B-16 model by tracing the semantic alignment of an image's internal representations on a layer-by-layer basis.

As a brief recall, the methodology involves projecting layer-specific activations into a shared semantic space using the neuron encoder $E_n$. We then compute the cosine similarity of these evolving representations against the text embeddings of both the ground-truth (GT) label and the label the model incorrectly predicted (the misclassified label). A divergence point, where the representation's similarity to the misclassified label surpasses that of the GT, marks the specific layer where the model's reasoning process falters.

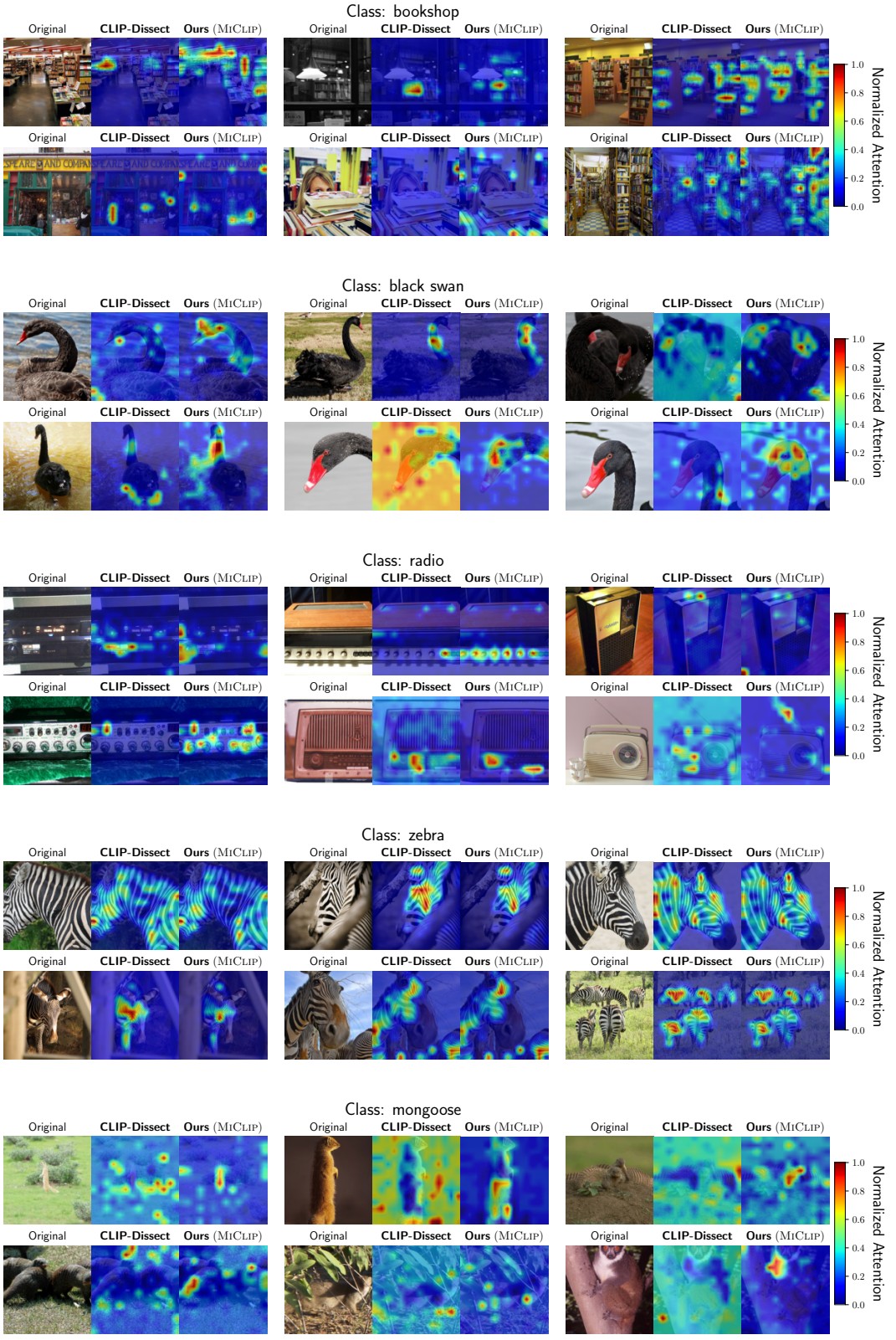

Figure 9: More results for visualizing the spatial grounding of top-5 localized features.

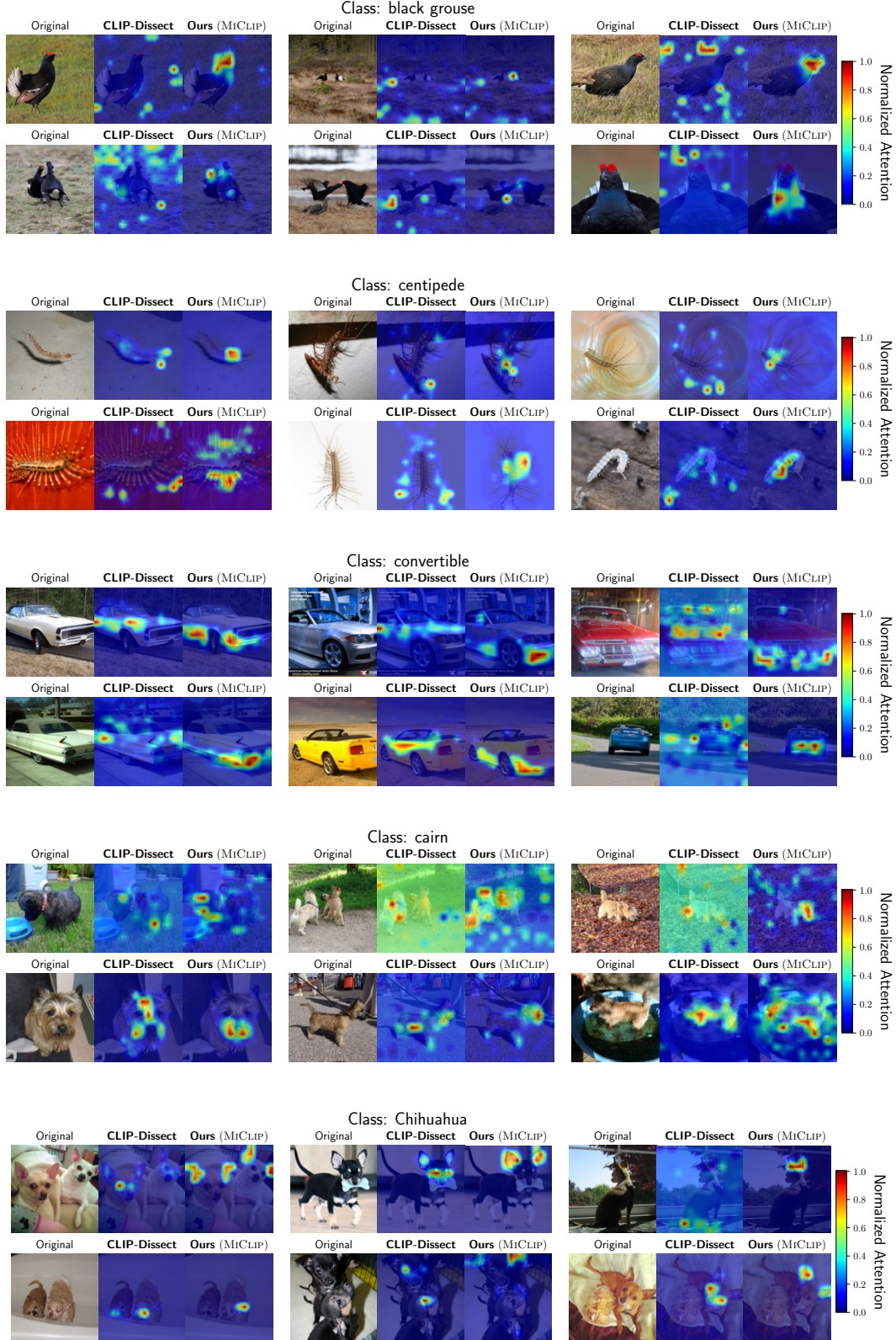

Figure 10: More results for visualizing the spatial grounding of top-5 localized features.

Figure 11 presents four additional examples of this flaw analysis. Each case demonstrates a distinct failure mode, yet our diagnostic method consistently identifies the critical layer where the semantic trajectory shifts away from the correct interpretation. These examples underscore the reliability of MICLIP as a tool for gaining precise insights into model failures, confirming that the crossover phenomenon is a general indicator of flawed visual reasoning.

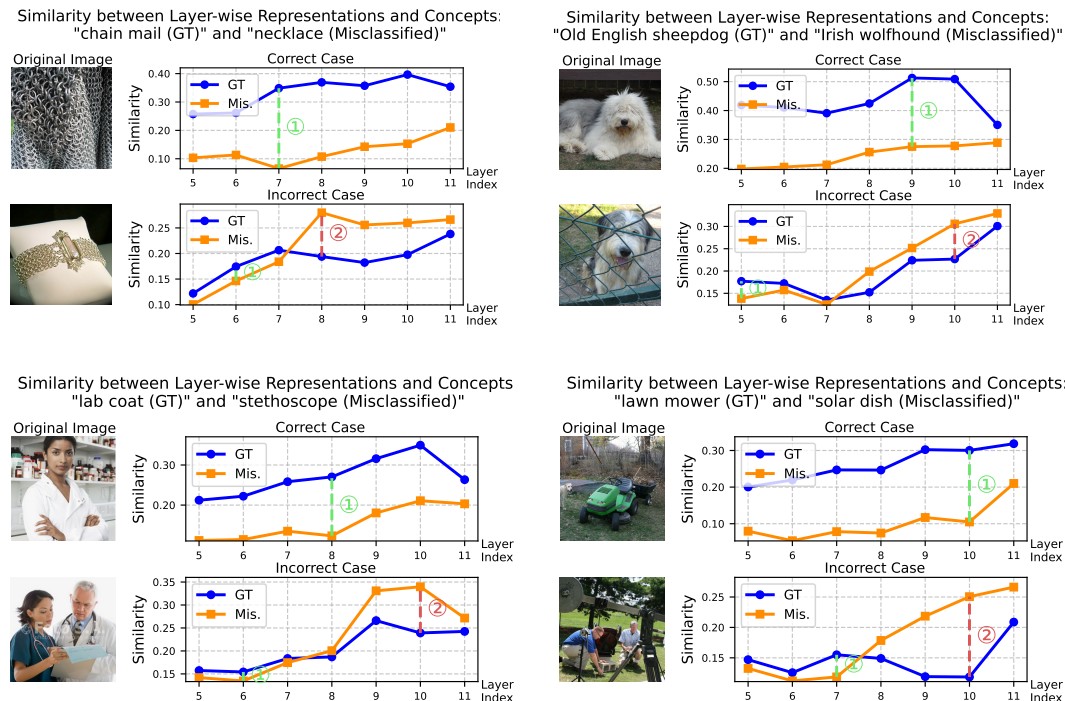

Figure 11: Additional results for flaw analysis. Each plot shows the cosine similarities between layer-wise representation embeddings and the text embeddings of the ground-truth (GT; blue) and the misclassified label (Mis.; orange). The layer where the GT similarity has the largest lead is marked by ①, while the critical failure point where the misclassified label's similarity overtakes the GT's is marked by ②.

### E.3 ADDITIONAL VISUALIZATION OF LOW LEVEL FEATURES

To further substantiate MICLIP's capability in identifying and localizing fundamental visual primitives, this appendix provides extended visualization results focusing on low-level features. While **Analysis 7** demonstrated the localization of color concepts (e.g., "green"), here we examine whether our method can effectively ground more complex low-level features, such as shapes, colors and textures, which are typically encoded in the early layers of the model.

We follow the identical visualization methodology described in **Analysis 7**. Specifically, we visualize the self-attention maps from the **3rd layer** of the CLIP/ViT-B-16 model.

Figure 12 and Figure 13 illustrates more spatial grounding of feature identified by MICLIP as corresponding to a **"shapes"**, **"colors"** or **"textures"** pattern. The results show that this feature consistently attends to structures, such as "red", "grid", or "round", across a diverse set of images. The results from object semantics verifies MICLIP's effectiveness in interpreting the model's behavior at different levels of granularity.

### E.4 USER STUDY OF LOCALIZED FEATURES

To quantitatively evaluate the quality of the interpretability heatmaps produced by our method, we conducted a user study to compare the ability of MICLIP and CLIP-Dissect to local-

ize relevant image regions. To ensure a fair comparison, we adhered to a strict blind testing protocol. An example of a question from our survey is shown in Figure 14.

**Participants and Recruitment.** We recruited participants with backgrounds in computer vision. None of the participants were authors of this paper, ensuring independence. We collected a total of 13 participants with 130 responses. The results, summarized in Table 6, show a clear user preference for MICLIP over CLIP-Dissect.

Table 6: User study results: User preference for explanations generated by each method.

| Method | User Preference (%) |
|---|---|
| CLIP-dissect | 23.08% |
| **MICLIP (Ours)** | **76.92%** |

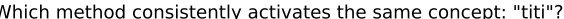

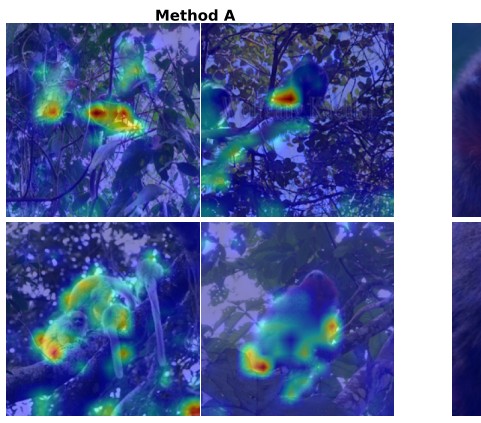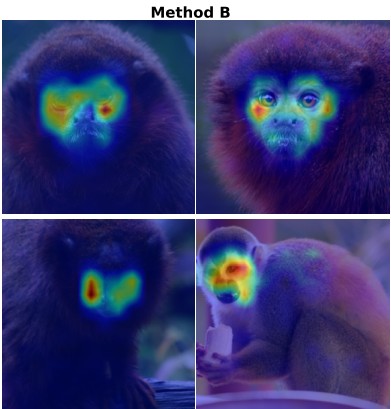

Figure 14: An example from our user study questionnaire. Participants were shown feature maps generated by two different methods (Method A and Method B, in random order) and were asked to select which one better explains the image's classification.

### E.5 VERIFYING MECHANISM LOCALIZATION VIA INTERVENTION ON LARGE VISION-LANGUAGE MODEL

To demonstrate the scalability of MICLIP, we extend our intervention analysis to the llava-hf/llava-v1.6-mistral-7b-hf model. We evaluate the model on the ImageNet-1k closed-set classification task (original accuracy 11.78%) using a specific prompt containing all 1,000 class labels (`{options_text}`).

We use the following prompt instructs the model: "What type of object is in this photo? Below is the exact list of choices. Each choice is enclosed in <> brackets for clarity. `{options_text}` Instructions: - Think step by step and then output the label. Choose EXACTLY ONE from the list above. Output the label inside the <> brackets. For example, if the answer is 'sea snake', output $< sea\ snake >$. Your response label must be a perfect and totally match to one of the labels."

Accuracy is calculated via exact string matching. The model's original classification accuracy is 11.78%. We perform interventions on SAE features within the **22nd layer** of the LLaVA vision tower. We compare MICLIP against CLIP-Dissect (Oikarinen & Weng, 2023) using enhancement ($\times 2$ scaling) and removal ($\times 0$ scaling) settings, averaged over three random seeds.

Table 7: Accuracy deviations ($\Delta Acc$ %) from enhancement and removal interventions on features on llava-hf/llava-v1.6-mistral-7b-hf model.

| Method | Enhancement ($\uparrow$) | Removal ($\downarrow$) |
|---|---|---|
| CLIP-dissect | 0.00 ($\pm$ 0.06) | 0.03 ($\pm$ 0.03) |
| **MICLIP (Ours)** | **1.11 ($\pm$ 0.01)** | **-1.87 ($\pm$ 0.05)** |

As shown in Table 7, MIClip effectively controls model behavior, achieving a **1.11%** gain in enhancement and a **1.87%** drop in removal. In contrast, the baseline CLIP-Dissect shows negligible impact, verifying that MIClip correctly identifies functionally relevant mechanisms in large-scale vision-language model.

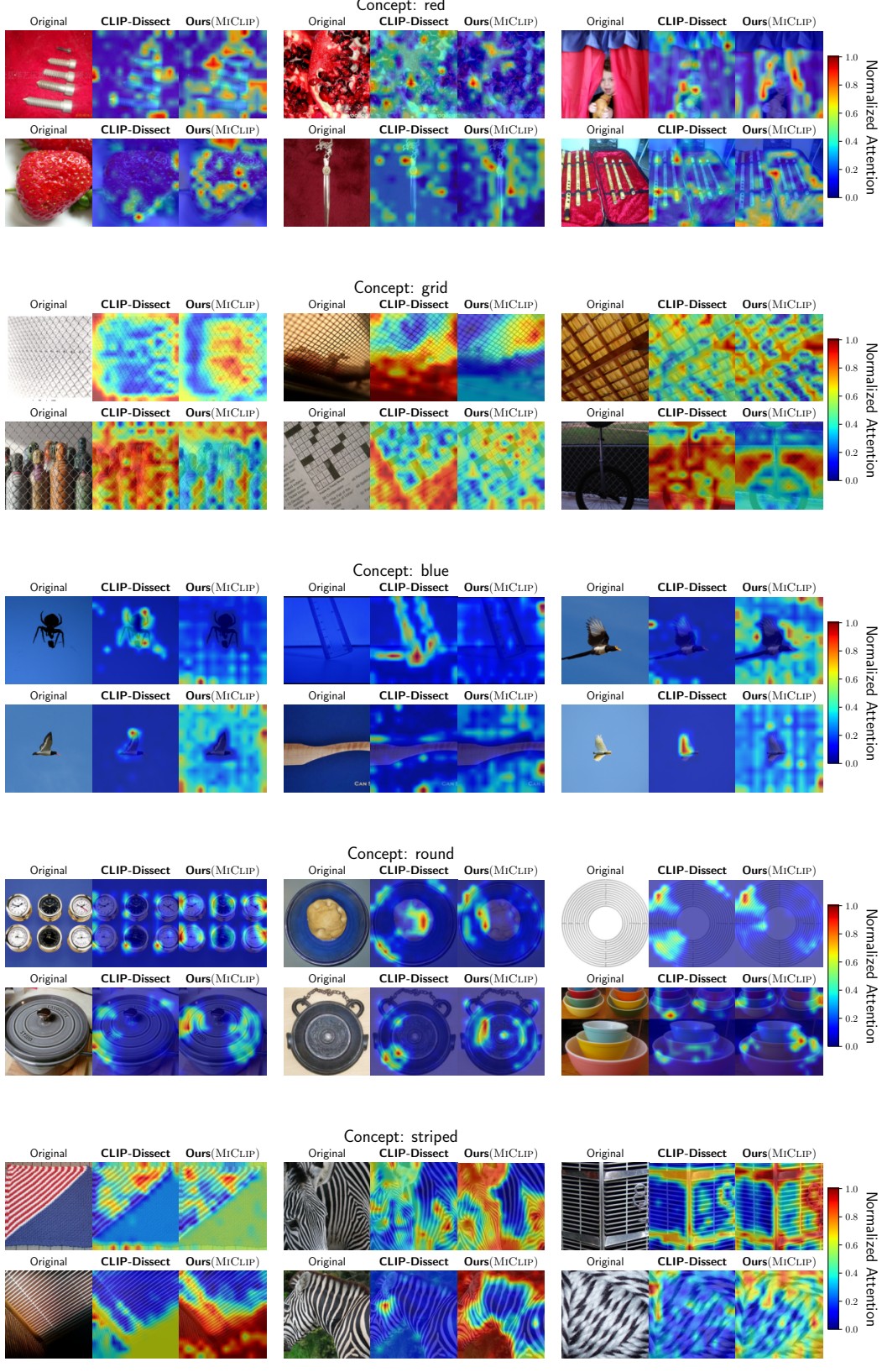

Figure 12: More results for visualizing the spatial grounding of color, shape and texture features.

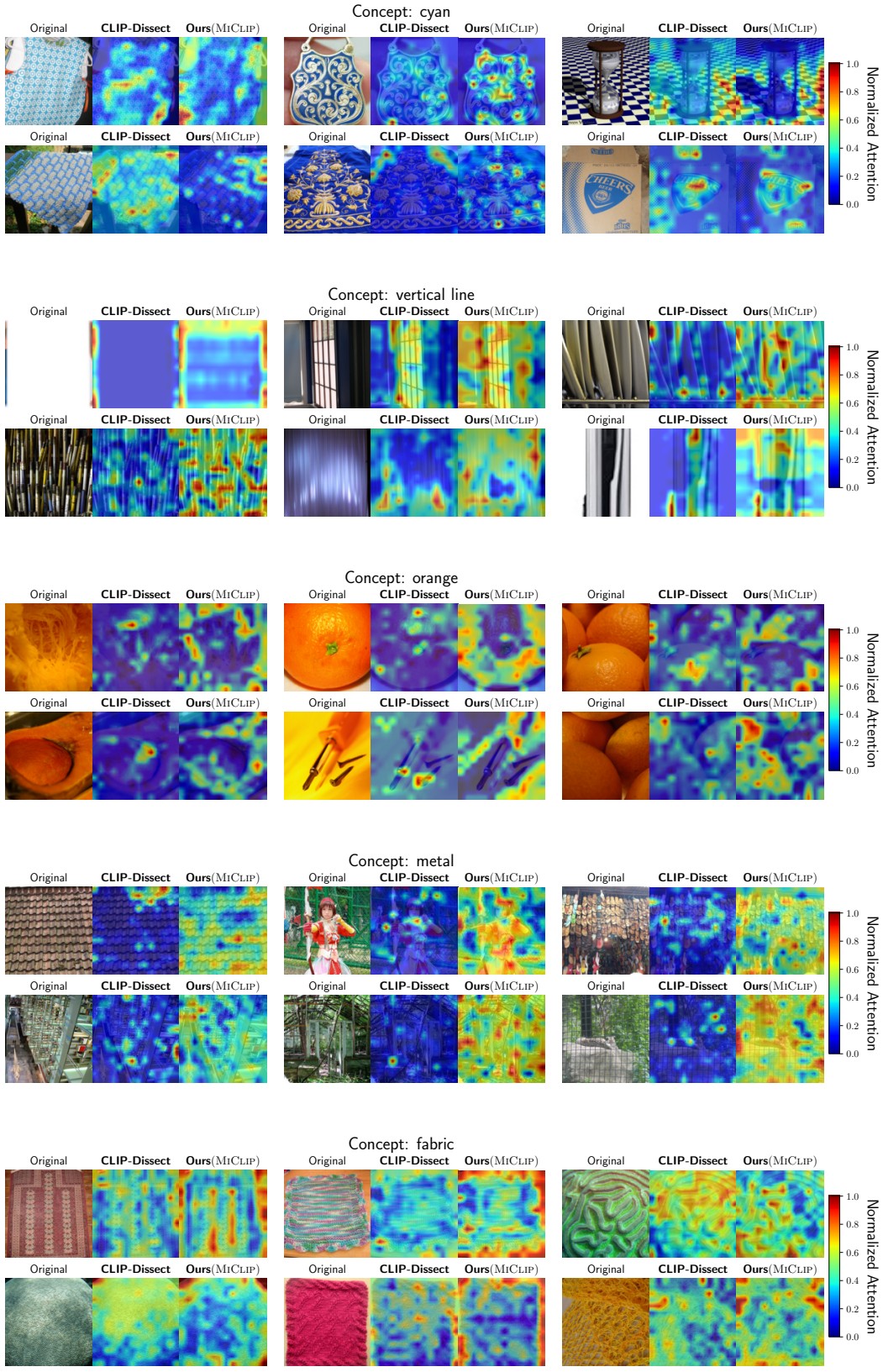

Figure 13: More results for visualizing the spatial grounding of color, shape and texture features.

