# OpenReview forum: "MICLIP: Learning to Interpret Representation in Vision Models"
_ICLR.cc/2026/Conference — ICLR 2026 Poster_

### Official Review · Reviewer_4V8N · 2025-10-27

**Soundness:** 3
**Presentation:** 3
**Contribution:** 4
**Rating:** 8
**Confidence:** 2

**Summary:**

The paper presents MICLIP, a framework that interprets internal units—either neurons or SAE-derived sparse features—by learning a lightweight encoder that maps their activations into CLIP’s semantic space. Training uses a symmetric InfoNCE objective that aligns each unit’s representation simultaneously with the input image embedding and the model’s predicted-label text embedding, replacing activation-magnitude heuristics with representation-level alignment. Once trained, the shared space enables concept→mechanism localization (finding units for a queried concept) and mechanism→concept description (annotating a chosen unit), and it naturally extends to SAE pseudo-activations to mitigate polysemanticity. The authors validate behavioral relevance through interventions that amplify or ablate selected units and consistently shift model outputs in the predicted direction, and they report strong results across ResNet/ViT/CLIP backbones with out-of-distribution generalization to texture concepts (DTD). Additional analyses, including layerwise error trajectories, demonstrate that MICLIP offers a practical, general recipe for unit-level interpretation without relying on unstable gradients or purely input-centric assumptions.

**Strengths:**

Gradient-free “causal” proxy via CLIP. Instead of using unstable gradients or handcrafted causal scores, the paper detours through CLIP to obtain a robust semantic target, then validates behaviorally via interventions.

Truly dual-anchor alignment (novel). Aligning activation→text and activation→image (not just label text) is a clean, original design that reduces input-centric bias and avoids the “activation-magnitude == importance” heuristic.

Works on SAE pseudo-activations. The description/localization pipeline applies neatly when only a single SAE feature is “turned on” via the decoder, which is practically valuable for polysemantic layers.

Thorough, convincing experiments. The paper hits standard metrics, reports state-of-the-art, includes interventions (↑ with amplify, ↓ with ablate), spans multiple backbones, and shows the method also works at OOD textures.

**Weaknesses:**

Not causal by design. The loss aligns to CLIP (image/text) semantics rather than optimizing a constraint tied to model logits or a formal mediator test.(Although this may be unstable) - so the causality holds between CLIP output and layer output, not between model output and layer output. The method demonstrates empirical causal relevance via interventions, but the mechanism is not guaranteed by the objective.

**Questions:**

Low-level features. Can you provide qualitative low-layer examples (e.g., “white horizontal line,” “45° diagonal line”) showing agreement between SAE top-activations and mechanism→concept retrieval? It would be a great aid to show MICLIP can effectively explain SAE features.

---

> ### Author Response · Authors · 2025-11-22
> **Rebuttal by Authors**
>
> Dear reviewer 4V8N, thank you for your appreciation and thoughtful questions! Following is our response.
>
> >  **W1 The method is not causal by design. Causality is not guaranteed by the objective.**
> >
> >  The loss aligns to CLIP (image/text) semantics rather than optimizing a constraint tied to model logits or a formal mediator test.(Although this may be unstable) - so the causality holds between CLIP output and layer output, not between model output and layer output. The method demonstrates empirical causal relevance via interventions, but the mechanism is not guaranteed by the objective.
>
> We thank the reviewer for pointing out the distinction between aligning hidden representations (i.e., layer output) to CLIP semantics and explicitly optimizing constraints tied to model logits or a formal mediation test.
>
> We deliberately choose CLIP-aligned semantic space because it provides a stable and semantically structured space shared across inputs, representations and the model outputs, and the mutual alignment has been conducted through our MICLIP method as described in Section 3.1 in our paper.
> Optimizing directly against model logits would overfit to task-specific decision geometry and may not provide a rich, interpretable notion of semantic interpretability of semantic similarity.
>
> However, we note that the layer representation is also aligned to the concept corresponding to the model's output (i.e., the predicted class with the highest logit) in CLIP semantic space, as shown in Eq. (2) in Section 3.1. Thus, the model output is explicitly incorporated.
>
> Moreover, directly using model logits as the alignment target leads to degenerate trivial solutions, e.g., hidden representations collapsing to directly predict logits, which undermines both interpretability and localization. Our goal is to extract semantically cohrerent mediators, for which CLIP serves as a well-validated semantic space.
>
> Although the alignment objective maps to CLIP space, the causal relevance to the model's own output is established through intervention experiments, as has been mentioned in your comments, where manipulating the discovered neurons or features changes the model's predictions in a controllable and meaningful manner.
>
> Our method MICLIP separates feature discovery (via semantic alignment) from causal validation (via interventions). While the objective itself aligns layer output to CLIP's semantic space, the causal link to the model's own output has been confirmed controlled interventions, demonstrating that the discovered features modulate the model's predictions. That is to say, our proposed CLIP-space alignment plays a role of the proxy for semantic mediation.
>
> We think that exploring more direct model logit-based constraints is indeed an interesting direction for future work, and we believe the present method has already provides empirical causal evidence tied to model outputs. Thanks again for your valuable points.
>
> > **Q1 Provide low-level features on low-layer examples**
> >
> > Low-level features. Can you provide qualitative low-layer examples (e.g., "white horizontal line," "45° diagonal line") showing agreement between SAE top-activations and mechanism→concept retrieval? It would be a great aid to show MICLIP can effectively explain SAE features.
>
> Thank you for the question. MICLIP does have the ability to capture low-level features, such as colors and textures.
>
> In Analysis 3, we adopted MICLIP to intervene target models on a low-level texture dataset DTD. The results are shown in Table 3, showing great intervention performance on low-level concepts.
>
> Further, per your review suggestions, we provide additional qualitative analysis on the low-level features.
> In **Section 4.7** of the revised version of paper, we adopt MICLIP to localize low-level concepts (e.g., colors, grids) on **SAE features** in the **3rd** layer, and then follow Analysis 5 in our paper to visualize the saliency map of located low-level SAE features.
>
> Compared with the baseline, MICLIP has demonstrated accurate localization ability of primitive concepts on the features learned in very low layer (Layer 3).
>
> For more results about MICLIP's performance on low-level features, you can refer to our General Response.

---

### Official Review · Reviewer_op3v · 2025-10-27

**Soundness:** 3
**Presentation:** 4
**Contribution:** 2
**Rating:** 4
**Confidence:** 4

**Summary:**

This paper proposes MiCLIP, a method that trains a CLIP neuron encoder to align with the visual and textual encoders of CLIP such that a neuron can be converted into an interpretable embeddings space (CLIP's embedding space) that can be queried via text. It aligns intermediate neuron activations of a target model (e.g., ResNet50) to the input image, and also adds the alignment of the activation values the predicted class labels for improved faithfulness. The method performs well compared to baselines.

**Strengths:**

- The paper is clearly written and easy to follow, it was an interesting read. I would like to thank the authors for the comprehensive appendix also.
- The authors tackle the "input-centric" paradigm (usually unfaithful to the mechanism of the model's decision-making process), and which often fail in cases of incorrect predictions (because they are explicitly trained on human-desired concepts). They do this by considering the model’s output decision.
- The proposed method can work with SAE features without actually training on those features (this means the SAE features are basis vectors of the dense features from the residual stream)
- The idea of representing a single neuron in the CLIP embedding space (via the Neuron encoder) is very interesting.
- The analysis is Figure 5 is interesting as it allows us to see the layer where the model fails.

**Weaknesses:**

- The paper assumes that the target layer they operate on, actually contains concepts they align to (here, the concepts are the predicted class labels). This does not really make sense to me. Different layers have different functions. For example, early layers represent low-level features such as textures, blobs, edges...etc. But the authors align them with high-level concepts (class labels). In other words, we never know the actual concepts that a target layer uses, so we cannot assume that these concepts are the predicted class labels. For me, this is a major weakness.
- The work of [R2] is very similar, it uses CLIP to learn interpretable embeddings (also sparse) which is exactly what the authors do. Although this work investigates CLIP, the general idea of learning a CLIP model to represent interpretable embeddings is the same.  Furthermore, [R3] also investigates the same idea of interpreting intermediate representations of models (including fine-grained neurons) with text by generating text in an autoregressive manner, and is trained on MILANNOTATIONS dataset [R4], a human-curated dataset for neuron annotation
- Why dont the authors include MILAN [R4] as a baseline? It is a popular baseline and should be reported.
- I am not convinced with the motivation of the paper. The claim in L52-53: "An increase in activation value does not necessarily imply the occurrence of the corresponding concept during inference. Conversely, even negative activations can positively influence the model’s
prediction of certain concepts." Is there any reference for this? How do the authors come up with such a claim? It has been well-established since 2013 that high activation values correspond to concepts (or parts of concepts). Potentially, they could be biases (e.g., dumbells and hand occur together so the "hand" concept will activate on dumbell images). However, high activation values still *drive* the propagation to other future layers and eventually the prediction. The authors repeatedly mention this problem all over the paper, but to me this problem is not well justified.
- Since the method aligns the neuron representation in the CLIP space, it therefore inherits CLIP's biases. For example, it may inherit the text-spotting bias in CLIP, even though this bias is not present in the target model.
- There is an evaluation method proposed for neuron annotation works [R1], have the authors considered this evaluation method? Right now, most of the evaluation involves interventions.
- In Table 1, why aren't the other baselines (Act-Values, Network Dissection and V-Interp) reported?
- In Table 2 I dont understand why dont authors report the overall accuracy of the model with enhancement? For example, ResNet-50 imagenet accuracy is around 76%. The the 5% boost, does that mean it is 81% ?
- The method seems to work less well for transformer models, according to Table 2.
- The description of Figure 5 is wrong. It mentions a CD player but it is actually a sea anemone and feather boa.
- Minor: In Table 1, what is the need of "Sig." ? Is it it fill space?

[R1] CoSy: Evaluating Textual Explanations of Neurons
[R2] STAIR: Learning Sparse Text and Image Representation in Grounded Tokens
[R3] DeViL: Decoding Vision features into Language
[R4] Natural Language Descriptions of Deep Visual Features

**Questions:**

At the current stage, there are many problems in the paper (see weaknesses). Therefore my decision will be borderline reject for now. My decision could potentially be adjusted in the rebuttal.

**Details Of Ethics Concerns:**

No issues

---

> ### Author Response · Authors · 2025-11-22
> **Rebuttal by Authors**
>
> Dear reviewer op3v, thank you for your appreciation of our paper's strength and valuable questions! Following is our response:
>
> > **W1 Assumption that the target layer operated on contains concepts MICLIP align to**
> >
> > The paper assumes that the target layer they operate on, actually contains concepts they align to (here, the concepts are the predicted class labels). This does not really make sense to me. Different layers have different functions. For example, early layers represent low-level features such as textures, blobs, edges...etc. But the authors align them with high-level concepts (class labels). In other words, we never know the actual concepts that a target layer uses, so we cannot assume that these concepts are the predicted class labels. For me, this is a major weakness.
>
> Thank you for your insightful question. We want to kindly correct that **MICLIP actually doesn't assume the target layer operated on contains the concepts MICLIP align to**. The reasons are as follows:
>
> **During training:**
>
> In reality, our method is designed to handle layers with varying levels of abstraction through two key mechanisms:
>
> - MICLIP aligns the layer activations not only with the predicted label $\mathcal{L}_{\mathrm{CLIP}}^{\text{output}}$ but also **directly with the input image $\mathcal{L}_{\mathrm{CLIP}}^{\text{input}}$**. Since the input image contains all primitive visual information (colors, edges, textures), early-layer activations (which process these primitives) have a strong correlation with the input image embedding in CLIP's space. If a layer lacks high-level semantic information, the optimization is naturally driven by the visual similarity term, aligning the layer with primitive visual concepts present in the CLIP space.
>
> - MICLIP leverages contrastive supervision, such as CLIP. The adopted contrastive objective **does not impose a hard assumption** that a layer *must* fully encode the label. Instead, contrastive learning maximizes the **mutual information** between the layer representation and the concepts. Crucially, maximizing mutual information encourages the model to capture features that are *correlated* with or *predictive* of the concept (e.g., "furry texture" correlates with the label "dog"), rather than forcing the layer's representation to be semantically identical to the high-level concept itself.
>
> These allows MICLIP to learn broad semantic correspondences without requiring exact grounding or objective-level alignment.
>
> **During Localization or Description (Utilization of MICLIP):**
>
> - When a layer's output indeed encodes a specific concept, the **relative similarity** measured by MICLIP between that representation and the concept becomes significantly higher than that for other layers (**not an absolute** value that indicates the presence of the concept). This is precisely what enables both "Concept-to-Mechanism Localization" and "Mechanism-to-Concept Description" in our framework, as discussed in Section 3.2 in the paper. Instead, we empirically verify through interventions that the discovered features indeed influence the model's predictions.
> - The **relative similarity** between a layer's output and a concept helps diagnose how the model processes information layer by layer, and reveals at which layer it begins to misinterpret inputs as other concepts, as shown in Section 4.6 of our paper.
>
> To further support MICLIP's capacity to recognize and intervene on low-level features, we have conducted specific experiments focusing on the model's early layers (e.g., Layer 3):
> - Qualitatively, in Section 4.7 of the revised paper, we visualize the spatial grounding of Sparse Autoencoder (SAE) features in Layer 3 of CLIP/ViT-B-16. These visualizations confirm MICLIP's accuracy in localizing specific primitive concept, such as colors, edges, and geometric shapes. The details of the experiments can be found in our **General Response**.
> - Quantitatively, Analysis 3 in the original paper demonstrates that MICLIP effectively identifies and intervenes on low-level texture concepts (using the DTD dataset) in intermediate layers (Layer 8, 9).
>
> We hope above points help resolve your concern.

---

> ### Author Response · Authors · 2025-11-22
> **Rebuttal by Authors**
>
> > **W2 More similar works like STAIR [R2] and DeViL [R3]**
> >
> > The work of [R2] is very similar, it uses CLIP to learn interpretable embeddings (also sparse) which is exactly what the authors do. Although this work investigates CLIP, the general idea of learning a CLIP model to represent interpretable embeddings is the same. Furthermore, [R3] also investigates the same idea of interpreting intermediate representations of models (including fine-grained neurons) with text by generating text in an autoregressive manner, and is trained on MILANNOTATIONS dataset [R4], a human-curated dataset for neuron annotation
>
> We acknowledge the similarity at first glance, but MICLIP is distinct from STAIR [R2] and DeViL [R3] from aspects of the target,  objective, and the scope of interpretability.
>
> Comparison with STAIR [R2]:
>
> - STAIR aims at learning sparse representations for downstream tasks' performance (e.g., image-text retrieval and classification). While MICLIP aims at learning to align the target model's internal space with human understandable features (both textual and visual) in CLIP embedding space.
> - STAIR enforces sparsity on the learned representation as a core training objective. MICLIP, on the other hand, only adopt the InfoNCE contrastive loss as the core training objective. The sparsity is only introduced to the target model through SAE, serving as an **optional external widget**, not a part of our method.
>
> Comparison with DeViL [R3]: While DeViL shares the goal of connecting internal components to human-understandable concepts, it's interpretability remains as saliency attribution discovery and textual caption generation, lacking integration into the model's reasoning path, which is MICLIP's core motivation and contribution.
>
> DeViL offers descriptive interpretability through neuron caption generation and text-based saliency attribution. MICLIP provides bidirectional alignment, enabling both description (mechanism-to-concept) and localization (concept-to-mechanism). Crucially, MICLIP's alignment framework allows us to intervene in model behavior given target concepts. It even enables diagnosing model's flawed behavior, as shown by Section 4.6 in the paper.
>
> We believe these papers share some similar points with MICLIP, but are works with distinct focuses.
>
> > **W3 Missing MILAN as baseline**
> >
> > Why dont the authors include MILAN [R4] as a baseline? It is a popular baseline and should be reported.
>
> Thank you for mentioning MILAN as a baseline.
>
> MILAN was not selected as a primary baseline because the models used for its training and validation are incompatible with our target models, specifically the ViT-B/16 and CLIP/ViT-B-16.
>
> However, considering MILAN's potential as a generative explanation method that could be generalized, we included it for comparison at the neuron level, following the same settings in Analysis 2 (interventions on ImageNet-1k concepts) and Analysis 3 (interventions on unseen DTD concepts) of the original paper. We measure the change in classification accuracy $\Delta Acc$, after applying either *enhancement* ($\times 2$ scaling) or *removal* ($\times 0$ scaling) to specific layers in the models. Since MILAN generates textual explanations for neurons, we utilized the CLIP/ViT-B-16 model to compute the similarity between MILAN's generated captions and the ImageNet-1k class labels. This allowed us to identify the most relevant neurons for intervention.
>
> **Table 1.1: $\Delta Acc$ (%) after Enhancement Intervention on Neurons Localized by ImageNet-1k Concepts (higher $\Delta Acc$ is better)**
> | Method | ResNet-50 | ViT-B/16 | CLIP |
> | :--- | :--- | :--- | :--- |
> | MILAN | -0.03 | -0.04 | -0.47 |
> | **MICLIP** | **5.32** | **0.18** | **1.10** |
>
> **Table 1.2: $\Delta Acc$ (%) after Removal Intervention on Neurons Localized by ImageNet-1k Concepts (lower $\Delta Acc$ is better)**
> | Method | ResNet-50 | ViT-B/16 | CLIP |
> | :--- | :--- | :--- | :--- |
> | MILAN | -0.16 | 0.00 | -0.17 |
> | **MICLIP** | **-17.24** | **-0.04** | **-1.50** |
>
> **Table 2: Intervention on neurons localized by DTD concepts ($\uparrow$: higher is better, $\downarrow$: lower is better)**
> | Method | **Enhancement $\Delta Acc$ (%) ($\uparrow$)** | Removal $\Delta Acc$ (%) ($\downarrow$) |
> | :--- | :--- | :--- |
> | MILAN | -0.17 | -0.31 |
> | **MICLIP** | **0.38** | **-0.91** |
>
> The results presented in Table 1.1 and Table 2 clearly indicate that the enhancement interventions guided by MILAN fail to improve the model's accuracy. In fact, they result in a slight decrease in performance across all three architectures. In contrast, our method (MICLIP) consistently demonstrates a significant and positive impact, successfully enhancing accuracy as intended.

---

> ### Author Response · Authors · 2025-11-22
> **Rebuttal by Authors**
>
> > **W4** I am not convinced with the motivation of the paper. The claim in L52-53: "An increase in activation value does not necessarily imply the occurrence of the corresponding concept during inference. Conversely, even negative activations can positively influence the model's prediction of certain concepts." Is there any reference for this? How do the authors come up with such a claim? It has been well-established since 2013 that high activation values correspond to concepts (or parts of concepts). Potentially, they could be biases (e.g., dumbells and hand occur together so the "hand" concept will activate on dumbell images). However, high activation values still drive the propagation to other future layers and eventually the prediction. The authors repeatedly mention this problem all over the paper, but to me this problem is not well justified.
>
> > **W4.1 Are there reference to L52-53?**
>
> Yes, there is. Previous work "How Important Is a Neuron?" [5] has mentioned similar results.
>
> > "Activation values for a ReLU based network are always positive. However, ReLU nodes can have positive or negative influence on the output depending on the upstream weights"
>
> This is especially salient if we focus on the following causal reasoning path of the model.
>
> > **W4.2 High activation values correspond to concepts are well-established since 2013**
>
> Thank you for your concern about our motivation.
>
> Higher activation values can indeed indicate a stronger correlation with the concept, but with some requirements on specific layer positions, such as post ReLU (e.g., DeconvNet Visualization[8] and CAM [9]). This becomes more salient if you consider model's following reasoning path, especially at the residual stream of transformer-based architectures, where there is no ReLU activation to guarantee a positive value.
>
> As illustrated in **W4.1**, a high activation value in intermediate layers have no theoretical gurantees to exert positive influence on following layers, not to mention the final output.
>
> The mentioned baselines [4], [6], [7] unanimously follow the activation-magnitude assumption, neglecting negatively activated components and only picking the positively activated ones. However, the neuron encoder in MICLIP will won't make assumptions on the magnitude of the hidden representations and directly align it through the causal relationship. This helps us to find more faithful and causally correlated concepts with model internals.
>
> We appreciate your comment. Our motivation is already justified, as we discussed in the text (L48 of the original paper).
>
> We hope above explanation can help you better understand our motivation and address your concern.

---

> ### Author Response · Authors · 2025-11-22
> **Rebuttal by Authors**
>
> > **W5 The method may inherit CLIP's bias which is not present in target model**
> >
> > Since the method aligns the neuron representation in the CLIP space, it therefore inherits CLIP's biases. For example, it may inherit the text-spotting bias in CLIP, even though this bias is not present in the target model.
>
> Thank you for your concern.
>
> Adopting the interpretable embedding space of CLIP is a common practice in this area.
> Our baseline CLIP-Dissect [6], like us, depend on CLIP embedding's structure to help align between concepts and model internals.
>
> However, we demonstrate empirically that CLIP's classification bias does not hinder MICLIP's performance.
>
> We partitioned the 1000 ImageNet-1k classes into two groups based on CLIP's zero-shot classification accuracy: the "Top 25%" group (250 classes) where CLIP performs best (average accuracy of 91.26%), and the "Bottom 25%" group (250 classes) where it performs worst (average accuracy of 30.39%).
> This split can illustrate some bias within the CLIP model on different semantic classes.
>
> Building upon our findings in Analysis 2 in the paper, we applied our neuron- and SAE-based intervention methods within the ViT-B/16 model for the two groups and measured the change in accuracy $\Delta Acc$. The results are presented in Tables 1 and 2 below.
>
> **Table 3: Intervention on Neurons**
> | Class Group | Enhancement $\Delta Acc$ (%) (↑) | Removal $\Delta Acc$ (%) (↓) |
> | :--- | :---: | :---: |
> | Top 25% | 0.14 | -0.16 |
> | Bottom 25% | **0.18** | **-0.44** |
>
>
> **Table 4: Intervention on SAE features**
> | Class Group | Enhancement $\Delta Acc$ (%) (↑) | Removal $\Delta Acc$ (%) (↓) |
> | :--- | :---: | :---: |
> | Top 25% | 2.89 | -29.10 |
> | Bottom 25% | **6.98** | **-31.46** |
>
> As shown in Tables 3 and 4, our method remains highly effective even on the "Bottom 25%" classes. Therefore, while CLIP has its own biases, our experimental results confirm that MICLIP does not simply inherit these biases.
>
> Moreover, we can also incorporate additional methods to alleviate or mitigate the influence of CLIP bias, such as "CLIP the bias" [2] and "CLIPood" [3], which have illustrated mitigating both the representation and data bias in CLIP model, as well as boost the OOD generalization ability of CLIP.
> These works are orthogonal to our method.
>
> Your point is interesting, and we believe it's worth further exploration as a promising future work. Thank you for your suggestions.
>
> > **W6 CoSy [R1] as evaluation method**
> >
> >  There is an evaluation method proposed for neuron annotation works [R1], have the authors considered this evaluation method? Right now, most of the evaluation involves interventions.
>
> We thank the reviewer for suggesting the CoSy evaluation framework [R1]. We have considered this method, but we believe it does not align with the specific goals of our paper for the following reasons:
>
> 1. CoSy checks "Input Correlation," while we check "Output Impact":
> CoSy evaluates a description by generating an image from that text and seeing if the neuron activates. This confirms that the neuron responds to the concept.
> However, our goal with MICLIP is to understand how the model uses that information to make a decision. Just because a neuron responds to a concept doesn't mean that concept actually changes the model's final prediction. CoSy only tests the first half of the process (Input to Neuron), whereas we need to verify the full process (Input to Neuron to Output).
> 2. Interventions provide stronger proof of reasoning:
> To truly understand the model's behavior, we need to know if a feature is necessary for the prediction. A neuron might activate strongly for a "simulated" image (as in CoSy), but still have no effect on the final classification.
> By using intervention metrics (changing the internal features and observing the result), we prove that the specific concept is not just present, but is actually causing the model's decision. This is a stricter and more reliable test for our objective of explaining the model's behavior.
>
> Due to this, we think this evaluation metric is not very consistent with our target and motivation.

---

> ### Author Response · Authors · 2025-11-22
> **Rebuttal by Authors**
>
> > **W7** In Table 1, why aren't the other baselines (Act-Values, Network Dissection and V-Interp) reported?
>
> Thank you for your question about the missing baselines in Table 1. The reason we omit the mentioned baselines are as follows:
>
> - Act-Values: the performance of Act-Values on the *ImageNet-1k dataset* is actually already reported in Table 1 of the original version of paper.
>     - For the Common-nk datasets, we exclude the results of Act-Values due to its demands on **class labels** on the dataset, which is *absent* in the Common-nk datasets as they are open-vocabulary words retrieved from the internet.
> - Network Dissection [5] and V-Interp [7]: Both Network Dissection and V-interp are incompatible with the setting of Analysis 1 (evaluation on the final layer) as these methods rely on the presence of a 2D spatial structure (e.g., intermediate convolutional layers), which does not exist in the final layer.
>
> > **W8 Overall accuracy in Table 2,3**
> >
> > In Table 2 I dont understand why dont authors report the overall accuracy of the model with enhancement? For example, ResNet-50 imagenet accuracy is around 76%. The the 5% boost, does that mean it is 81% ?
>
> Yes, your interpretation is correct. The $\Delta Acc$ shown in Table 2 and Table 3 is the difference between the accuracy after intervention and the classification accuracy of the original model, as reported by the target models.
>
> We demonstrate the accuracy change as it is a direct reflection and depiction of how effective the intervention is to deviate the classification accuracy towards the desired direction.
>
> We agree that reporting the original accuracy improves clarity. We have revised accordingly in the new version of our paper.
>
> > **W9 Work less for transformer architecture in Table 2**
> >
> > The method seems to work less well for transformer models, according to Table 2.
>
> The experiment in Table is conducted under two settings: intervention and SAE features and model neurons.
>
> - Intervention on SAE features: **MICLIP performs much better on transformer-based model** (ViT-B/16 and CLIP) compared with ResNet-50.
> - Intervening neurons, the performance on transformer-based models is not as good as that on the ResNet-50.
>
> Our explanation is that the difference is likely resulted by *Polysemanticity* [10] inherent in transformer-based architectures, rather than the defection in MICLIP. The relative more narrow layer in ViT (768 compared with 1024 in ResNet) may exacerbate this polysemanticity, making the target neuron more challenging to be effectively intervened upon.
>
> In conclusion, we believe the superior performance observed with SAE features demonstrates that MICLIP successfully mitigates the difficulty posed by raw polysemantic neurons.
>
> > **W10 Text figure mismatch for Figure 5**
> >
> > The description of Figure 5 is wrong. It mentions a CD player but it is actually a sea anemone and feather boa.
>
> We apologize for this mistake. This has been solved in the revised version of our paper (Section 4.6).
>
> > **W11 What is the need for "Sig." in Table 1**
> >
> > Minor: In Table 1, what is the need of "Sig." ? Is it it fill space?
>
> Thank you for your question. By writing "Sig.", we are showing the result of significance test, not for filling space. We apologize for the ambiguity and have moved it to the text part in the revised version of our paper (Section 4.2).
>
> > **Reference:**
> >
> > [1] Rao, Sukrut, et al. "Discover-then-name: Task-agnostic concept bottlenecks via automated concept discovery." European Conference on Computer Vision. Cham: Springer Nature Switzerland, 2024.
> >
> > [2] Alabdulmohsin, Ibrahim, et al. "CLIP the Bias: How Useful Is Balancing Data in Multimodal Learning?" The Twelfth International Conference on Learning Representations, 2024.
> >
> > [3] Shu, Yang, et al. "CLIPood: Generalizing CLIP to Out-of-Distributions." International Conference on Machine Learning, 2023.
> >
> > [4] Bau, David, et al. "Network Dissection: Quantifying Interpretability of Deep Visual Representations." Computer Vision and Pattern Recognition, 2017.
> >
> > [5] Dhamdhere, Kedar, et al. "How Important Is a Neuron." International Conference on Learning Representations, 2019.
> >
> > [6] Oikarinen, et al. "CLIP-Dissect: Automatic Description of Neuron Representations in Deep Vision Networks." The Eleventh International Conference on Learning Representations, 2023.
> >
> > [7] Xhang, Kaichen, et al. "Large multi-modal models can interpret features in large multi-modal models." Proceedings of the IEEE/CVF International Conference on Computer Vision. 2025.
> >
> > [8] Zeiler, Matthew D., et al. "Visualizing and Understanding Convolutional Networks." European Conference on Computer Vision. Cham: Springer, 2014.
> >
> > [9] Zhou, Bolei, et al. "Learning Deep Features for Discriminative Localization." Computer Vision and Pattern Recognition, 2016.
> >
> > [10] Elhage, et al., "Toy Models of Superposition", Transformer Circuits Thread, 2022.

---

### Official Review · Reviewer_E7Mu · 2025-10-30

**Soundness:** 2
**Presentation:** 2
**Contribution:** 3
**Rating:** 2
**Confidence:** 4

**Summary:**

This paper proposes a mechanistic interpretability approach for vision models, by projecting model internal representations to CLIP embedding space. The authors train an encoder to project model activations to CLIP embedding space via two contrastive losses: (1) between the projection and the CLIP embedding of the model output and (2) between the projection and the CLIP embedding of the image itself. The idea is to then project one-hot activations (resembling individual neurons) to CLIP space and thus learn which concept a unit represents. One can then conduct causal interventions on models, by modulating the units relevant for a concept. The authors validate their method by applying it to the final classification layer of various models and showing that the projections are close to CLIP / Mpnet embeddings of the ground-truth labels.

**Strengths:**

- The motivation of the work is very good, both the field's reliance on the activation-magnitude assumption and the fact that existing approaches focus on input-centric explanations too much is valid.
- The work is well-placed in the related literature.
- The proposed method is applied to multiple models of different architectures, and compared against multiple reasonable baselines.

**Weaknesses:**

- The implicit assumption behind this work, which is never addressed in the text, is that CLIP embedding space is somehow interpretable. To what extent this is true is hard to say, but my intuition is that while the method may work for the kinds of object-level representations one finds in late layers, it likely won't transfer to earlier layers which detect more primitive features.
- The clarity of the writing could be improved. For example, from figure 2 one might think that the target model is trained or fine-tuned (because while the clip-encoders are show to be frozen, the target model does not) and the paper generally spends more time making claims than backing them up (eg, line 198). The "Common-Nk" datasets in table 1 are never explained, I have to assume they are subsets of COCO?
- One technical problem that I see is that $E_n(a_i \cdot e^{(i)})$ is probably OOD for the encoder. The $a_i$ would have to be post-relu activations of very sparse layers for training on real activation vectors to generalize to these base vectors. Investigating vectors of unit length is also questionable because this assumes that the scaling of neuron activations is somewhat standardized, which might not be the case.
- The validation of the approach in table 1 is in my opinion insufficient, because the final classification layer is special w.r.t. the two issues outlined above: It represents high-level semantic concepts, and is likely much sparser than other layers.
- The reporting of significance in table 1 is superfluous: At 100k test images, even very small differences are significant. I would save the space and write in the caption that all differences are significant.
- In principle, I like the analysis in table 2, but I would have done it differently: You are currently not evaluating the specificity of interventions. When you are removing units responsible for a concept, you are probably not only hurting performance on the target class, but also on other classes. The relevant performance metric should not just be the absolute delta in classification accuracy for the target class, but the relative drop in accuracy for the target class relative to other classes. For example, it's possible that the -17% average accuracy delta for the target class you find in ResNet-50 is not different from all other classes, which could also be dropping by 17%. Higher specificity would mean that the target class accuracy drops by 17% while the average accuracy of all other classes drops by only 1%, for example.
- I thus don't think findings 2 and 3 are sufficiently supported by evidence (in Table 3 the CIs overlap and the absolute deltas are very small).
- Minor point, but in section 4.6 the example in the text doesn't match the example in figure 5.
- At the very end of section 4.6 you claim that your method points out "where the model's view shifts and why it fails". The first part is reasonable, the latter should be removed: We have no idea why the similarity to the wrong label surpasses that to GT, we can just observe that it happens.

Overall, I am quite unimpressed with the paper. I initially liked the pitch of overcoming the activation-magnitude assumption, but I don't think the approach convincingly shows that it achieves any of its objectives. It is a general issue with the field of interpretability that evaluation is hard and usually not properly done, but I find these evaluations particularly insufficient.

**Questions:**

- I assume you keep the target model frozen and only train the encoder, but just to make sure: Which parts of the training pipeline are frozen?

---

> ### Author Response · Authors · 2025-11-22
> **Rebuttal by Authors**
>
> Dear reviewer E7Mu, thanks for the comments and following is our response.
>
> > **W1.1 Implicit assumption that CLIP embedding space is interpretable**
> >
> > The implicit assumption behind this work, which is never addressed in the text, is that CLIP embedding space is somehow interpretable. To what extent this is true is hard to say, but my intuition is that while the method may work for the kinds of object-level representations one finds in late layers, it likely won't transfer to earlier layers which detect more primitive features.
>
> Thank you for pointing out this assumption.
>
> The interpretability of CLIP's embedding space has already been discussed by some previous works, and is widely utilized by works in our area.
>
> "Discover-then-Name" [1], has proven that the embedding space of CLIP is highly structured and interpretable. It applies Sparse Autoencoder to the embedding space, and found thousands of features that corresponds to human-understandable concepts. Inspired by this, we adopted CLIP's interpretable embedding space to help analyze other vision models.
>
> Adopting the interpretable embedding space of CLIP is a common practice in this area.
> Our baseline CLIP-Dissect [2], like us, depends on CLIP embedding's structure to help align between human-understandable semantic concepts and model internals.
>
> We have a more detailed discussion how we utilize the interpretability of CLIP's embedding space in the response to **W3.1**.
>
> Hope this can help address your concerns.
>
> > **W1.2 The method may work for latter layers, but hard to transfer to earlier layers with more primitive features**
>
> Our experimental results can prove that MICLIP is able to generalize to different level of concepts from different layers.
>
> While early layers process low-level signals (e.g., textures, edges), MICLIP is designed to capture them because it aligns model internals with the *input image*, not just the high-level output class labels. Since the input image contains all primitive visual information, MICLIP can effectively bridge early-layer activations to primitive concepts.
>
> We include detailed analysis of the experiments that support MICLIP's generalization performance across different granularities of concepts in the **General Response**, and summarize as below:
>
> - Quantitative: Analysis 3 in our paper demonstrates effective intervention on low-level textures (DTD dataset) not seen during training.
> - Qualitative: In the revised Section 4.7, we visualize the spatial grounding of Layer 3 features, confirming that MICLIP precisely localizes primitive concepts like colors and shapes in early layers.
>
> > **W2 Writing clarity. Is target model trained and "Common-Nk" not clear**
> >
> > The clarity of the writing could be improved. For example, from figure 2 one might think that the target model is trained or fine-tuned (because while the clip-encoders are show to be frozen, the target model does not) and the paper generally spends more time making claims than backing them up (eg, line 198). The "Common-Nk" datasets in table 1 are never explained, I have to assume they are subsets of COCO?
>
> Thank you for the suggestions about writing clarity.
>
> - The target model is **NOT** trained or finetuned. The only trainable part is the neuron encoder in our MICLIP.
> - The "Common-Nk" (N = 3k / 10k / 20k) concept set used in our experiments follows exactly the same construction as the open-vocabulary concept set introduced in CLIP-Dissect [2]. It consists of the **N most frequent English words** extracted from a widely used public word-frequency list.
>
> We thank the reviewer for pointing this out. We have updated Figure 2 and Section 4.2 in the revised version to clarify this point.

---

> ### Author Response · Authors · 2025-11-22
> **Rebuttal by Authors**
>
> > **W3.1 $E_n(a_i \cdot e^{(i)})$ can be OOD, and $a_i$ would have to be post-relu activations of very sparse layers for training**
> >
> > One technical problem that I see is that $E_n(a_i \cdot e^{(i)})$ is probably OOD for the encoder. The $a_i$ would have to be post-relu activations of very sparse layers for training on real activation vectors to generalize to these base vectors.
>
> We appreciate your technical concern regarding the Out-of-Distribution (OOD) nature of the single-neuron case, $E_n(a_i \cdot e^{(i)})$, during the testing phase.
>
> We clarify that the design (linear projection) of our neuron encoder $\mathrm{E_n}$ **guarantees the training and testing phases are mathematically consistent**, which prevents the single-neuron projection from being OOD.
>
> #### 1. Recalling MICLIP's training objective
>
> MICLIP's mechanism-concept alignment process is formulated as a contrastive learning problem.
> Given a set of neuron activatinos $\mathcal{A}$ and the corresponding set of predicted concepts $ \\{ \\hat{c} _ j \\} _ { j = 1 } ^ N$, the objective is to align their embedding through the alignment loss:
>
>
> $$
> \\mathcal{L} _{ \\mathrm{alignment} } = \\mathcal{L} _{ \\mathrm{CLIP} } ^ { \\text{output} } \\left( \\mathrm{E} _{ \\mathrm{n} } ( \\mathcal{A} ; \\theta _{ \\text{n} } ) , \\mathrm{E} _{ \\mathrm{c} } ( \\{ \\hat{c} _ j \\} _{ j=1 } ^ N ) \\right) + \\mathcal{L} _{ \\mathrm{CLIP} } ^ { \\text{input} } \\left( \\mathrm{E} _{ \\mathrm{n} } ( \\mathcal{A} ; \\theta _{ \\text{n} } ) , \\mathrm{E} _{ \\mathrm{i} } ( \\mathcal{X} ) \\right)
> $$
>
>
> The core of this objective is the symmetric InfoNCE loss  which maximizes the similarity of positive pairs. Focusing on the neuron-image loss, $\mathcal{L}_{\mathrm{CLIP}}^{\text{input}}$ is a bidirectional sum of the InfoNCE losses. For a positive pair $(\mathbf{n}, \mathbf{x})$, where $\mathbf{n} = \mathrm{E_n}(\mathbf{a})$, the loss in one direction (neuron to image) is:
>
> $$
> \ell_{\text{InfoNCE}} = - \log \frac{\exp (\mathrm{sim}(\mathbf{n}, \mathbf{x})/\tau)}{\sum_{\mathbf{x}'} \exp \left( \mathrm{sim}(\mathbf{n}, \mathbf{x}')/\tau \right)}
> $$
>
> The summation in the denominator is taken over all image embeddings $\mathbf{x}'$ in the batch. The image-to-neuron loss is similar and we can derive $\mathcal{L}^{\text{output}}_{\text{CLIP}}$ likewise.
>
> #### 2. In-Domain Consistency Ensured by Linearity of $\mathrm{E_n}$
>
> **The key to resolving the OOD concern lies in the linearity of $\mathrm{E_{n}}$, which ensures that optimizing for the activation vector $\mathbf{a}$ implicitly optimizes for its constituent basis vectors.**
>
> Our neuron encoder $\mathrm{E_n}$ is designed as a linear projection. The activation vector is $\mathbf{a} = \sum_{i=1}^n a_i \cdot e^{(i)} \in \mathbb{R}^n$, where $\\{e^{(i)}\\}$ is the standard basis. Due to linearity, the output $\mathbf{n}$ decomposes:
>
> $$\mathbf{n} = \mathrm{E_n}(\mathbf{a}) = \mathrm{E_n} \left(\sum_i a_i e^{(i)}\right) = \sum_i a_i \underbrace{\mathrm{E_n}(e^{(i)})}_{\text{Projected Basis Vector}}$$
>
> Inspired by previous work [1], by assuming the CLIP image embedding space is spanned by semantic basis vectors $\\{v_l\\}$ (i.e., $\mathbf{x}=\sum_l k_l \cdot v_l$), the similarity function becomes a weighted sum of component-wise alignments:
>
> $$\mathrm{sim}(\mathbf{n}, \mathbf{x}) = \mathrm{sim}\left(\sum_i a_i \mathrm{E_n}(e^{(i)}), \sum_l k_l v_l\right) = \sum_{i,l} a_i k_l \cdot \underbrace{\mathrm{sim}(\mathrm{E_n}(e^{(i)}), v_l)}_{\text{Single Neuron-Semantic Align}}$$
>
> This demonstrates that the training objective *directly aligns the fundamental components of the activation vector* (the projected basis vectors $\mathrm{E_n}(e^{(i)})$) with the semantic basis vectors $\\{v_l\\}$ of the CLIP space.
>
> **Therefore, the single-neuron projection, $\mathrm{E_n}(e^{(i)})$, is explicitly optimized during the contrastive training process.**
>
> This mathematical decomposition demonstrates that the training objective directly aligns the fundamental components of the activation vector (the projected basis vectors $\mathrm{E_n}(e^{(i)})$) with the semantic basis vectors $\\{v_l\\}$ of the CLIP space.
>
> This conclusion is maintained even when decomposing the activation $\mathbf{a}$ using Sparse Autoencoder (SAE) features $\\{f_i\\}$, as the training process aligns these interpretable features with CLIP's semantic features.
>
> We hope this clarifies how the linearity of $\mathrm{E_n}$ ensures consistency between the training of the full activation vector $\mathbf{a}$ and the testing or analysis of its single-neuron components $e^{(i)}$.

---

> ### Author Response · Authors · 2025-11-22
> **Rebuttal by Authors**
>
> > **W3.2 Investigating vectors of unit length is questionable, which assumes that the scaling of neuron activations is standardized**
> >
> > Investigating vectors of unit length is also questionable because this assumes that the scaling of neuron activations is somewhat standardized, which might not be the case.
>
> Thank you for raising this point about the scaling of neuron activations.
>
> We clarify that our investigation of unit-length vectors does **not** rely on the assumption that neuron activations are standardized.
>
> This is because all the similarity calculated during MICLIP's training, localization, and description, **are all cosine similarity**:
>
> $$
> \\mathrm{sim}(x,y) = \\frac{\\langle x,y \\rangle}{\\|x\\| \\cdot \\|y\\|} .
> $$
>
> Cosine similarity is **inherently independent of vector magnitude**. It only measures the angular alignment between the two vectors.
>
> Therefore, since we only measure the embedding similarity through cosine similarity **both in the training stage and the test stage**, we ensure that the method is invariant to the magnitude of the neuron activation $\mathbf{a}$.
>
> Our focus remains on the directional relationship of the embeddings.
> We believe this represents the feature content captured by the neuron, irrelevant of its original scale. This makes the method robust to potential non-standardized scaling across different neurons or layers.
>
> > **W4 Table 1. Final classification layer is special since it represents high-level semantic concepts and is much spareser than other layer**
> >
> > The validation of the approach in table 1 is in my opinion insufficient, because the final classification layer is special w.r.t. the two issues outlined above: It represents high-level semantic concepts, and is likely much sparser than other layers.
>
> Thank you for the question about Analysis 1. We conduct the experiment on the final classification layer for following reasons:
>
> 1. The last layer is the only layer with Grouth Truth.
> 2. For direct and fair comparison with the baselines. Analysis 1 directly follows the setting of Table 1 in CLIP-Dissect [2].
>
> We agree that the final layer may be special with high-level semantic concepts and is much sparser. Actually, **MICLIP is able to generalize to low-level concepts at early layers**.
>
> As shown in the **General response**, we conducted qualitative proof in the revised Section 4.7. We visualize the spatial grounding of **3rd Layer** features with low-level concepts (e.g., colors, shapes and stripes), confirming that MICLIP precisely localizes primitive concepts like colors and shapes.
>
> We also have provided validation on intermediate layers in the original paper. **Analysis 5** verifies the spatial grounding of features on the **8th layer** via attention maps, while **Analysis 2 and 3** demonstrate the causal effectiveness of our method on intermediate layers (e.g., the **10th layer**) through intervention experiments on both seen and unseen concepts.
>
> We hope this experiment help solve your concern.
>
> > **W5 Superious Significance report in Table 1**
> >
> > The reporting of significance in table 1 is superfluous: At 100k test images, even very small differences are significant. I would save the space and write in the caption that all differences are significant.
>
> Thanks for your valuable suggestion!
>
> Since the numerical result values of the baselines and MICLIP are of close margin (though on large-scale evaluation datasets), we thus conduct the significance test to make Anlaysis 1 comprehensive and rigoruous. We have followed your advice and revised the paper accordingly (Table 1 in Section 4.2).

---

> ### Author Response · Authors · 2025-11-22
> **Rebuttal by Authors**
>
> > **W6 Table 2: Need accuracy delta of other classes for specificity**
> >
> > In principle, I like the analysis in table 2, but I would have done it differently: You are currently not evaluating the specificity of interventions. When you are removing units responsible for a concept, you are probably not only hurting performance on the target class, but also on other classes. The relevant performance metric should not just be the absolute delta in classification accuracy for the target class, but the relative drop in accuracy for the target class relative to other classes. For example, it's possible that the -17% average accuracy delta for the target class you find in ResNet-50 is not different from all other classes, which could also be dropping by 17%. Higher specificity would mean that the target class accuracy drops by 17% while the average accuracy of all other classes drops by only 1%, for example.
>
> Thank you for your detailed advice. We agree that evaluating the impact on non-targeted classes is essential to verify the *specificity* of our interventions.
>
> The primary reason we reported target-class accuracy in the main paper is the computational prohibition. A full specificity evaluation requires running inference on the entire validation set for every single class intervention, inflating the computational cost by approximately $1,000\times$ (evaluating on 1000 classes) than the current setting.
>
> Per your suggestion, we conducted a verification experiment on ViT-B/16 following the same intervention setting of Analysis 2 in the original paper. We report the "Average Accuracy on Non-Targeted Classes" (Other Acc) alongside the "Target Class Accuracy" (Target Acc) after intervention.
>
> **Table 2: Target annd Non-Targeted Classes Accuracy after intervention on ViT-B/16**
> | Method | Intervention | Target Acc (%) | Other Acc (%) |
> | :--- | :--- | :---: | :---: |
> | **Original** | - | 80.32 | 80.32 |
> | **MICLIP (Neuron)** | Enhancement | **80.49** (+0.17) | 80.29 (-0.03) |
> | | Removal | **80.15** (-0.17) | 80.31 (-0.01) |
> | **MICLIP (SAE)** | Enhancement | **85.90** (+5.58) | 80.29 (-0.03) |
> | | Removal | **48.21** (-32.11) | 80.31 (-0.01) |
>
> As shown in Table 2, while our method causes significant performance shifts on the *Target Class* (especially on SAE features), the average accuracy of *Other Classes* remains remarkably stable (fluctuations $\le 0.03$%).
>
> > **W7 Insufficient support in Table 2,3 for Finding 2,3. Table 3: CI overlap and the absolut edeltas are very small**
> >
> > I thus don't think findings 2 and 3 are sufficiently supported by evidence (in Table 3 the CIs overlap and the absolute deltas are very small).
>
> Thank you for raising concerns about the result of Table 2,3 of the original paper.
>
> **Regarding the "small deltas":** We respectfully clarify that the "small deltas" are expected when intervening on single neurons in large models like CLIP/ViT-B-16. This is because an individual neuron often polysemantically encodes multiple distinct concepts.
>
> **Regarding CI overlap in Table 3 of the original paper:** While the reported Confidence Intervals (CI) for certain settings in the main results may exhibit overlap with the baseline, it is crucial to examine the robustness and consistency of the underlying selection process across varying conditions.
>
> **Table 3: Detailed Accuracy Deviations after Enhancement Experiment on Neurons in Analysis 3** ($\uparrow$: higher is better; "Seed" refers to the random seed for data splitting in both CLIP-Dissect and MICLIP methods)
> | Method | Seed = 0 $\Delta Acc$ ($\uparrow$) | Seed = 42 $\Delta Acc$ ($\uparrow$) | Seed = 333 $\Delta Acc$ ($\uparrow$)|
> | :--- | :---: | :---: | :---: |
> | **CLIP-Dissect** | -0.19% | +0.18% | +0.02% |
> | **MICLIP** | **+0.22%** | **+0.45%** | **+0.48%** |
>
> As shown in Table 3, CLIP-Dissect exhibits high variance and can identify "wrong" neurons that *degrade* performance (Seed = 0). In contrast, **MICLIP consistently aligns with the correct direction** across all seeds.
>
> We hope these improvements will make the result more salient and help solve your concern.
>
> > **W8 Text figure mismatch for Fig. 5**
> >
> >  Minor point, but in section 4.6 the example in the text doesn't match the example in figure 5.
>
> Thank you for pointing this out. We apologize for the typo. It is now corrected in the revised version of the paper.

---

> ### Author Response · Authors · 2025-11-22
> **Rebuttal by Authors**
>
> > **W9 Overclaim that MICLIP can explain "why model fails"**
> >
> >  At the very end of section 4.6 you claim that your method points out "where the model's view shifts and why it fails". The first part is reasonable, the latter should be removed: We have no idea why the similarity to the wrong label surpasses that to GT, we can just observe that it happens.
>
> We thank the reviewer for this precise observation. We agree that claiming to explain fully "why the model fails" implies a causal mechanism that might be overstated based solely on similarity plots.
>
> While we cannot fully explain the neuron-level causality of why the shift happens, MICLIP offers important diagnostic hints by finding out where the error emerges.
>
> 1. As shown in Figure 5 (and Appendix), we observe a consistent pattern in misclassification cases. The similarity to the Ground Truth (GT) and the Misclassified label exhibits a sudden divergence (e.g., around layers 7 and 8 in the reported example).
> 2. This "crossover point" indicates exactly at which depth the model begins to prioritize features associated with the wrong class (e.g., "feather boa") over the correct one (e.g., "sea anemone").
>
> This localization allows researchers to investigate specific layers for bias or feature entanglement, rather than searching the entire model.
>
> Following your advice, we have updated the text in Section 4.6 to reflect this clearer, more rigorous interpretation.
>
> > **Q1: Which parts of the training pipeline are frozen?**
> >
> > I assume you keep the target model frozen and only train the encoder, but just to make sure: Which parts of the training pipeline are frozen?
>
> Both the CLIP encoders and the target model are frozen. The only trainable component is the neuron encoder.
>
> We have revised Figure 1 to make this design obvious and mentioned the frozen target model in the caption.
>
> > **Reference**
> >
> > [1] Rao, Sukrut, et al. "Discover-then-name: Task-agnostic concept bottlenecks via automated concept discovery." European Conference on Computer Vision. Cham: Springer Nature Switzerland, 2024.
> >
> > [2] Oikarinen, et al. "CLIP-Dissect: Automatic Description of Neuron Representations in Deep Vision Networks." The Eleventh International Conference on Learning Representations, 2023.

---

### Official Review · Reviewer_sbm3 · 2025-11-01

**Soundness:** 2
**Presentation:** 2
**Contribution:** 2
**Rating:** 4
**Confidence:** 3

**Summary:**

This paper introduces MiCLIP, a novel framework for mechanistic interpretability of vision models that aligns internal representation units (neurons or sparse features) with human-understandable concepts in CLIP's semantic space. The key innovation is using contrastive learning to create a shared embedding space that connects model internals with both input images and output predictions, moving beyond the traditional "activation-magnitude assumption." MiCLIP enables bidirectional interpretation (concept-to-mechanism localization and mechanism-to-concept description) and supports model steering through targeted interventions. Extensive experiments on ResNet-50, ViT-B/16, and CLIP/ViT-B-16 demonstrate superior performance in interpretation accuracy, intervention effectiveness, and generalization to unseen concepts compared to established baselines.

**Strengths:**

1. **Novel and Well-Motivated Approach:** The dual input-output grounding through contrastive learning represents a significant advancement over existing input-centric interpretability methods. Moving beyond the activation-magnitude assumption addresses a fundamental limitation in current mechanistic interpretability research.

2. **Comprehensive Technical Framework:** The method provides a unified approach that works with both neurons and sparse autoencoder features, offering flexibility across different granularities of model internals. The integration with k-SAE for feature disentanglement is particularly valuable for addressing polysemanticity.

3. **Thorough and Multi-faceted Evaluation:** The paper presents an extensive experimental evaluation covering:
   - Quantitative interpretation accuracy across multiple models and concept sets
   - Intervention studies demonstrating precise model control
   - Generalization to unseen concepts and datasets
   - Semantic geometry analysis and attention map visualizations
   - Failure diagnosis through layer-wise semantic trajectory analysis

4. **Practical Utility:** The framework enables meaningful model steering and provides insights into flawed reasoning processes, making it valuable for both understanding and improving vision models.

**Weaknesses:**

1. **Dependence on CLIP's Semantic Space:** The interpretability power is inherently limited by CLIP's semantic coverage and potential biases. A more detailed discussion of these limitations and how they might affect interpretation fidelity would strengthen the paper.
2. **Architectural Scope:** While the method is evaluated on several discriminative vision architectures, its applicability to generative models or larger vision-language models remains unexplored. Some discussion of potential challenges in scaling to these domains would be valuable.
3.**Computational Cost Analysis:** Although Table 5 provides FLOPs comparison, a more comprehensive analysis of training time, memory requirements, and scalability to larger models would help practitioners assess practical deployment.
4. **Lack of Human Evaluation for Interpretability:** While the paper provides extensive automated evaluation, it lacks human studies to validate whether the discovered concept-mechanism alignments are actually meaningful and useful to human users. Automated metrics like CLIP score don't necessarily correlate with human-judged interpretability quality. The approaches of ACE (Towards Automatic Concept-based Explanations) and CRAFT(CRAFT: Concept Recursive Activation FacTorization for Explainability) to measure the interpretability quality could be considered

**Questions:**

See weakness

---

> ### Author Response · Authors · 2025-11-22
> **Rebuttal by Authors**
>
> Dear reviewer sbm3, thank you for your appreciation and comments. Following is our response.
>
> > **W1 Dependence on CLIP's Semantic Space**
> >
> > The interpretability power is inherently limited by CLIP's semantic coverage and potential biases. A more detailed discussion of these limitations and how they might affect interpretation fidelity would strengthen the paper.
>
> We appreciate the reviewer's insightful comment regarding the potential limitations imposed by CLIP's semantic coverage and biases. We acknowledge that, like other state-of-the-art methods (e.g., CLIP-Dissect [5]), MICLIP leverages CLIP's embedding structure to align internal mechanisms with interpretable concepts. This dependency on large, pre-trained models (such as CLIP) is a fundamental, shared challenge in this area of research.
>
> We have added such discussion in the revised version of our paper (Section 5, Conclusion), which is shown in blue. Thank you for your advice.
>
> However, we demonstrate empirically that CLIP's classification bias does not hamper MICLIP's ability to identify and intervene on relevant features.
>
> To verify this, we partitioned the 1000 ImageNet-1k classes into two groups based on CLIP's zero-shot classification accuracy: the "Top 25%" group (250 classes) where CLIP performs best (average accuracy of 91.26%), and the "Bottom 25%" group (250 classes) where it performs worst (average accuracy of 30.39%).
> This split can illustrate some bias within the CLIP model on different semantic classes.
> Building upon our findings in Analysis 2 in the paper, we applied our neuron- and SAE-based intervention methods within the ViT-B/16 model for the two groups and measured the change in accuracy $\Delta Acc$. The results are presented in Tables 1 and 2 below.
>
> **Table 1: Intervention on Neurons ($\uparrow$: higher is better, $\downarrow$: lower is better;)**
> | Class Group | Enhancement $\Delta Acc$ (%) ($\uparrow$) | Removal $\Delta Acc$ (%) ($\uparrow$) |
> | :--- | :---: | :---: |
> | Top 25% | 0.14 | -0.16 |
> | Bottom 25% | **0.18** | **-0.44** |
>
> **Table 2: Intervention on SAE features ($\uparrow$: higher is better, $\downarrow$: lower is better;)**
> | Class Group | Enhancement $\Delta Acc$ (%) ($\uparrow$) | Removal $\Delta Acc$ (%) ($\uparrow$) |
> | :--- | :---: | :---: |
> | Top 25% | 2.89 | -29.10 |
> | Bottom 25% | **6.98** | **-31.46** |
>
> As shown in the tables, MICLIP performs *better* (larger $\Delta Acc$) on the "Bottom 25%" classes. The results clearly indicate that CLIP's initial bias on a class does not hinder the performance of our intervention.
>
> Moreover, we can also incorporate additional methods to alleviate or mitigate the influence of CLIP bias, such as "CLIP the bias" [1] and "CLIPood" [2], which have illustrated mitigating both the representation and data bias in CLIP model, as well as boost the OOD generalization ability of CLIP.
> These works are orthogonal to our method.
>
> Your point is interesting, and we believe it's worth further exploration as a promising future work. Thank you for your suggestions.
>
> > **W2 Architectural Scope**
> >
> > While the method is evaluated on several discriminative vision architectures, its applicability to generative models or larger vision-language models remains unexplored. Some discussion of potential challenges in scaling to these domains would be valuable.
>
> Thank you for pointing this out.
>
> MICLIP can be easily transferred and adapted to different models besides discriminative models, such as vision-language models (VLMs).
>
> To demonstrate this, we applied MICLIP to a large vision-language model, specifically llava-hf/llava-v1.6-mistral-7b-hf [6]. The model was evaluated on a closed-set ImageNet-1K classification task, where its original accuracy was 11.78%. Following the intervention setting defined by Finding 2 in our original paper, we intervened on the 22nd layer of the vision tower. We focused this experiment specifically on SAE features to evaluate MICLIP's effectiveness in VLM, contrasting its performance against the strong baseline, CLIP-Dissect [5]. We report the average change in accuracy ($\Delta Acc$) over three random seeds below.
>
> **Table 3: Intervention on SAE features**
> | Method | Enhancement $\Delta Acc$ (%) (↑) | Removal $\Delta Acc$ (%) (↓) |
> | :--- | :---: | :---: |
> | CLIP-Dissect | 0.00 (± 0.06) | 0.03 (± 0.03) |
> | **MICLIP (Ours)** | **1.11 (± 0.01)** | **-1.87 (± 0.05)** |
>
> As shown in Table 3, while CLIP-Dissect fails to identify effective SAE features (yielding a **+0.03%** $\Delta Acc$ for removal), MICLIP demonstrates precise control, achieving a **1.11% accuracy gain** in the enhancement setting and a **1.87% drop** in the removal setting. This shows that MICLIP is applicable to not only discriminative models like ViT, but also to different models such as larger VLMs. We will add discussion about this in the paper.

---

> ### Author Response · Authors · 2025-11-22
> **Rebuttal by Authors**
>
> > **W3 Computational Cost Analysis**
> >
> > Although Table 5 provides FLOPs comparison, a more comprehensive analysis of training time, memory requirements, and scalability to larger models would help practitioners assess practical deployment.
>
> MICLIP demonstrates notable computational efficiency, which is critical for scalable concept analysis. This efficiency is reflected not only in reduced FLOPs but also in key metrics such as *time cost* and *memory requirements*. We benchmarked these resource demands against baseline methods, and the results are presented in Table 4.
>
> Table 4 details the resource requirements for concept analysis, specifically the mechanism-concept alignment and concept-to-mechanism tasks in Section 3 of the original paper, comparing **Training Time** (neuron encoder cost), **Inference Time** (concept description generation), **VRAM Memory** (GPU cost), and **Storage Memory** (disk space for cached activations and final concepts). All benchmarks were conducted using a single NVIDIA V100 GPU with a batch size of 1024. To ensure a fair comparison, the reported results represent the aggregated duration over 100,000 images in ImageNet-1k dataset under a unified experimental setting. All benchmarks were conducted on a ViT-B-16 backbone using a single NVIDIA V100 GPU with a batch size of 1024.
>
> **Table 4: Comparison of Computational Cost on ViT-B-16.**
> | Method            | Training Time          | Inference Time       | VRAM Memory    | Storage Memory |
> | :---------------- | :--------------------- | :------------------- | :------------- | :------------- |
> | CLIP-Dissect  | N/A                    | ~30 minutes          | ~16 GB | ~9.7 GB        |
> | Act-Values  | N/A                    | ~30 minutes          | ~16 GB | ~97 MB        |
> | **MICLIP (Ours)** | ~30 minutes            | **~2 minutes**       | ~16 GB | **~94 MB**     |
>
> We believe this shows the efficiency of MICLIP compared with the baselines.
>
> We have updated the Table 5 in Appendix C.2 of the original paper with these additional results. Thank you for your advice!
>
> > **W4 Lack of Human Evaluation for Interpretability**
> >
> > While the paper provides extensive automated evaluation, it lacks human studies to validate whether the discovered concept-mechanism alignments are actually meaningful and useful to human users. Automated metrics like CLIP score don't necessarily correlate with human-judged interpretability quality. The approaches of ACE (Towards Automatic Concept-based Explanations) and CRAFT(CRAFT: Concept Recursive Activation FacTorization for Explainability) to measure the interpretability quality could be considered.
>
> Thank you for your suggestion to include human evaluation to validate the alignments learned by MICLIP!
>
> We followed your advice and include a user study to measure the interpretability quality of MICLIP.
>
> To evaluate the experiment, we recruited 13 independent participants with backgrounds in computer vision. We present participants with heatmaps of localized features generated by MICLIP and the baseline, CLIP-Dissect.  Participants were presented with paired heatmaps for the same image and were asked to select the one that better explained the ground-truth label.  To ensure a fair comparison, the study was conducted in a blinded manner with randomized display orders. Further details regarding the experimental protocol are provided in **Appendix E.4** of the revised paper.
>
> The results from 13 valid responses are summarized below, showing a strong preference for our MICLIP method.
>
> | Method       | User Preference (%) |
> | :----------- | :------------------ |
> | CLIP-Dissect | 23.08%              |
> | **MICLIP**   | **76.92%**          |
>
>
> > **Reference**
> >
> > [1] Alabdulmohsin, Ibrahim, et al. "CLIP the Bias: How Useful Is Balancing Data in Multimodal Learning?" The Twelfth International Conference on Learning Representations, 2024.
> >
> > [2] Shu, Yang, et al. "CLIPood: Generalizing CLIP to Out-of-Distributions." International Conference on Machine Learning, 2023.
> >
> > [3] Ghorbani, Amirata, et al. "Towards Automatic Concept-Based Explanations." Advances in Neural Information Processing Systems, 2019
> >
> > [4] Fel, Thomas, et al. "CRAFT: Concept Recursive Activation FacTorization for Explainability." Proceedings of the IEEE/CVF Conference on Computer Vision and Pattern Recognition (CVPR), 2023, pp. 2711–21.
> >
> > [5] Oikarinen, et al. "CLIP-Dissect: Automatic Description of Neuron Representations in Deep Vision Networks." The Eleventh International Conference on Learning Representations, 2023.
> >
> > [6] Liu, Haotian, et al. "Improved Baselines with Visual Instruction Tuning." Proceedings of the IEEE/CVF Conference on Computer Vision and Pattern Recognition (CVPR), 2024, pp. 26286–26296.

---

### Author Response · Authors · 2025-11-22
**General Response**

We appreciate all the reviewers' insightful and valuable feedback!

We are greatly encouraged by the positive comments of reviewers, e.g.,
* "The dual input-output grounding through contrastive learning represents a significant advancement over existing input-centric interpretability methods," supported by a "thorough and multi-faceted evaluation."  (Reviewer sbm3)
* "The motivation of the work is very good," tackling "the field's reliance on the activation-magnitude assumption," and the method is "applied to multiple models" with "reasonable baselines." (Reviewer E7Mu)
* "The paper is clearly written and easy to follow," effectively "tackles the 'input-centric' paradigm," and proposes the "interesting" idea of representing single neurons in the CLIP embedding space. (Reviewer op3v)
* The "truly dual-anchor alignment" is a "clean, original design" that avoids heuristic biases, backed by "thorough, convincing experiments" and "reports state-of-the-art." (Reviewer 4V8N)

Following the valuable suggestions, we have uploaded a revised version of the paper for reference, with all the revisions shown in blue.

**A common and important question** raised by the reviewers is whether MICLIP can capture low-level primitive features (e.g., colors, textures, shapes) in the early layers of the target model:

- "but my intuition is that while the method may work for the kinds of object-level representations one finds in late layers, it likely won't transfer to earlier layers which detect more primitive features." (Reviewer E7Mu)
- "For example, early layers represent low-level features such as textures, blobs, edges...etc. But the authors align them with high-level concepts (class labels)." (Reviewer op3v)
- "Can you provide qualitative low-layer examples (e.g., "white horizontal line," "45° diagonal line") showing agreement between SAE top-activations and mechanism→concept retrieval? It would be a great aid to show MICLIP can effectively explain SAE features." (Reviewer 4V8N)

**Here, we provide a unified answer to this question**: MICLIP is indeed capable of recognizing low-level features in early layers and can precisely localize these concepts without finetuning or retraining the model. The experimental evidence is provided as follows:

- **Quantitative Results:** Analysis 3 (Section 4.3) in original version of the paper illustrates the generalization ability of MICLIP to low-level concepts from the *Describe Texture Dataset* (DTD), which was *not* used during MICLIP's training.
The results show that MICLIP can effectively intervene on the target model using the features it identifies corresponding to low-level concepts in DTD.

- **Qualitative Results:** In addition to the quantitative results mentioned above, we have included qualitative visualizations in **Section 4.7** of the revised paper. Adopting the attention map visualization method from Analysis 5, we illustrate the spatial grounding of SAE features in the **3rd** layer of the target model identified by MICLIP. These visualizations demonstrate that the identified features are highly relevant to primitive concepts such as colors, shapes, and textures.

We believe the above results collectively demonstrate MICLIP's strong performance in locating primitive concepts, illustrating its effectiveness on interpreting different granularity of concepts across different layers of the target model.

---

### Author Response · Authors · 2025-11-27
**Appreciation for Reviewers‘ Comments; Welcoming Further Discussion**

Dear Reviewers,

We hope this message finds you well.

We posted a detailed response a few days ago. As the discussion period is progressing, we wanted to gently check if our response and the additional results have sufficiently clarified your questions. We remain fully available to provide further explanations.

Thank you again for your time and constructive feedback.

Best regards,

Authors

---

### Author Response · Authors · 2025-12-03
**Author Final Remarks to Area Chairs  (1/2)**

Dear Area Chair,

We are writing to provide a concise executive summary of our rebuttal progress and the consensus among reviewers prior to the recent AC reassignment.

During the discussion period, we gave serious consideration to the valuable comments from all reviewers and offered detailed responses, including comprehensive new experiments and theoretical clarifications. However, the recent data leakage incident occurred before the reviewers could reply to our updates or revise their scores.

We understand that the review scores may have been reverted to their pre-rebuttal state. However, during the discussion period, we made significant progress in addressing the reviewers' concerns.

To assist in your assessment, we have summarized our key contributions and resolutions to the reviews' concerns as follows.

## Summary of Contributions

To assist in your evaluation, we briefly recap the core contributions of our paper:

**1. Challenging the "Activation-Magnitude" Assumption:** Many related work trying to align model internals with human understandable concepts often contain an implicit "Activation-Magnitude" assumption, that a higher activation value of a unit indicates a stronger presence of its corresponding concept in model's reasoning path. This assumption hasn't been strictly supported, and our MICLIP jump out of the assumption to adopt the semantic similarity in an embedding space to align model internals with concepts.

**2. An Early Exploration to Interpret Models with Dual-anchored Input-output Grounding Paradigm:** Existing approaches to interpret model internals mainly focus on input-centric explanations. MICLIP makes the advancement to align model internal representations with both the input and the output of the model.

**3. A Comprehensive Alignment Framework Able to Discover Different Granularity of Concepts in Different Representation Units, Different model's Different layers:** MICLIP is a comprehensive framework compatible with both neurons and Sparse Autoencoder (SAE) features and align them with both high-level objective concepts (e.g., kit fox) and low-level primitive concepts from both latter or early layers (e.g., colors, textures, grids). It is also compatible to different models with different architectures (e.g., ViT, ResNet, VLMs).

**4. Thorough and multi-facted validations ranging from quantitative intervention to qualitative human study and heat visualization:** We validate our MICLIP through a comprehensive suite of experiments, including quantitative causal interventions, qualitative spatial heatmap grounding, and a blinded human evaluation, all prove MICLIP's outperforming effectiveness.

## Review of Rebuttal Process

### Part 1: Consensus on Strengths

We are encouraged that reviewers unanimously appreciated MICLIP's core contributions:

**1. Breaking the "Activation-Magnitude" Assumption for more faithful concept grounding:**

Most reviewers validated our motivation to move beyond the *"activation-magnitude" assumption* (larger activation $\implies$ stronger presence of the corresponding concept in model's reasoning process). **This assumption lacks rigorous support and often oversimplifies complex model behaviors.**

In contrast, MICLIP introduces a learning-based paradigm to bypass this, employing contrastive learning to directly align model internals with concepts for faithful grounding.

**2. Extend from Input-centric Methods to Dual-anchored Input-Output Causal Interpretability:**

MICLIP establishes a dual-anchored framework grounding internals in both input visual semantics (raw signals) and output textual concepts (prediction semantics). By bridging input stimulation and output reasoning, it captures the complete processing procedure for holistic explanations.

All the reviewers validated this as a significant paradigm shift and contribution.

**3. Expanding Architectural Scope (VLMs) and Benchmarking Rigor** (Reviewers sbm3, op3v)

Reviewers raised concerns regarding the generality of MICLIP across different model families (e.g., Vision-Language Models) and the sufficiency of our compared baselines. We addressed these by expanding our evaluation:

- **New Target Model (VLM):** For Reviewer sbm3's request, we applied MICLIP to large VLM, specifically llava-hf/llava-v1.6-mistral-7b-hf [1] following the settings of Analysis 2 in paper.
    - **Results:** MICLIP demonstrated precise control over the VLM, significantly outperforming the baseline CLIP-Dissect.
- **Human Evaluation:** For Reviewer sbm3's request, we conducted a blinded user study building upon the visulization of localized features.
    - **Results:** Users preferred MICLIP over the baseline CLIP-Dissect in 76.92% of cases for interpretability quality.
- **Missing Baseline (MILAN):** Addressing Reviewer op3v's concern about missing MILAN as a baseline, we compared MICLIP with MILAN under the settings of Analysis 2 and 3 in paper.
    - **Results:** MICLIP outperforms MILAN significantly.

---

> ### Author Response · Authors · 2025-12-03
> **Author Final Remarks to Area Chairs (2/2)**
>
> **4. Extensive and convincing evaluation**
>
> We validated MICLIP through a multi-faceted set of experiments:
> - **Quantitative**: We employed causal intervention experiments to demonstrate MICLIP’s ability to precisely steer model behavior, proving the causal relevance between identified features and the model's predictions.
> - **Qualitative**: New heatmap visualizations verified accurate localization of diverse concept granularities, ensuring spatial fidelity.
> - **Human-Aligned Interpretation**: To ensure semantic accuracy, we conducted a blinded human evaluation, where MICLIP was preferred over the baseline in 76.92% of cases.
>
> ### Part 2: Reviewers' Questions and Our Resolution
>
> During the rebuttal period, we addressed all reviewer concerns with new experiments and theoretical clarifications. We added these revisions in our paper, marked by blue fonts.
>
> **1. Generalization to Low-level Primitive Features** (E7Mu, op3v, 4V8N)
>
> The reviewers raised some questions about MICLIP's generalization to low-level primitive features and lower layers of the target models. We have proved that MICLIP **does generalize to different granularities of concepts on different layers** through both intuitive and experimental evidence.
> - **Intuition:**
>     - The input-output-anchored design and CLIP's interpretable embedding space enable MICLIP to adapt to low-level primitive concepts.
> - **Empirical:** We provided both qualitative and quantitative experiments to support the generalization ability.
>     - Qualitative: Visualizations in Section 4.7 of the revised paper show that MICLIP precisely localizes primitive concepts (e.g., colors, shapes, grids) in early layers (Layer 3 of ViT-B).
>     - Quantitative: Re-emphasis of Analysis 3 confirms that MICLIP effectively intervenes on the model using low-level texture concepts (DTD dataset) unseen during training, proving robust transferability to primitive features.
>
> **2. Dependence on CLIP's Bias and Performance** (sbm3, op3v)
>
> Reviewers questioned if MICLIP inherits CLIP's potential biases. We demonstrated that MICLIP's steering ability is orthogonal to CLIP's zero-shot accuracy:
> - **Experiment:** Following Analysis 2, we split ImageNet-1k classes into "Top 25%" and "Bottom 25%" groups based on CLIP's accuracy.
> - **Result:** MICLIP's intervention remains robust and even performs better on the "Bottom 25%" classes.
>
> This empirically proves MICLIP's effectiveness is not hampered by CLIP's classification bias. (See Response to sbm3, Table 1 & 2)
>
> **3. Expanding Architectural Scope (VLMs) and Benchmarking Rigor** (Reviewers sbm3, op3v)
>
> We addressed concerns regarding generality (extend to VLMs) and missing baselines by expanding our evaluation:
> - **New Target Model (VLM):** We applied MICLIP to LLaVA-v1.6-Mistral-7b [1], following the settings of Analysis 2.
>     - **Results:** MICLIP demonstrated precise control over the VLM, significantly outperforming the baseline.
> - **Human Evaluation:** We conducted a blinded user study on feature visualizations.
>     - **Results:** Users preferred MICLIP over CLIP-Dissect in 76.92% of cases.
> - **Missing Baseline (MILAN):** We compared MICLIP against MILAN under Analysis 2 & 3 settings.
>     - **Results:** MICLIP outperforms MILAN significantly.
>
> **4. Clarification on Methodological Validity: Training Consistency (E7Mu)**
>
> Reviewer E7Mu questioned if single-neuron projection is OOD, as training adopted dense vectors.
>
> We solved this question by showing mathematical consistency between training and testing:
> - *Training*: Due to $\mathrm{E_n}$'s linearity, the training loss on dense vectors decomposes into the sum of pairwise similarities between constituent units and concepts.
> - *Testing* utilizes this exact optimized mapping. Thus, projecting a single unit is in-domain by construction, as the training phase explicitly aligns these individual components with the semantic space.
>
> For rigorous derivation, please refer to the response to Reviewer E7Mu (W3.1).
>
> ## Conclusion
>
> We have rigorously strengthened the paper during the rebuttal phase. With the integration of the LLaVA experiment, a blinded User Study, qualitative Heatmap Visualization, the MILAN baseline, and a theoretical proof of consistency, we believe we have comprehensively resolved the reviewers' initial concerns and significantly elevated the quality of the paper.
>
> While it is unfortunate that the recent incident prevented the reviewers from engaging with these updates, we believe the added experiments speak for themselves regarding the paper's quality. We respectfully request that you consider these substantial improvements in your final assessment.
>
> Finally, we deeply appreciate your time and leadership in navigating this review process under such unprecedented and challenging circumstances.
>
> Sincerely,
>
> Authors
>
> > **Reference**
> >
> > [1] Liu, Haotian, et al. "Improved Baselines with Visual Instruction Tuning."
> >
> > [2] Gao, Leo, et al. "Scaling and Evaluating Sparse Autoencoders."

---

### Meta-Review · Area_Chair_LETo · 2026-01-06

**Summary:**

This paper received 4 reviews which are quite mixed: Reviewer 4V8N champions the paper, Reviewer E7Mu manifested not being that impressed, Reviewers sbm3 and op3v are more borderline-low side

Overall the idea of moving away from activation-magnitude / input focused practices was well received. Similarly for the conducted evaluation which included multiple architectures and  baselines.

During the rebuttal/discussion phase, the the concerns raised in  the reviewers were, to a good extent, addressed. During this phase, the paper was extended with a user study, the inclusion of large vision-language model (llava-hf/llava-v1.6mistral-7b-hf [6]) and other experiments/tests suggested by the reviewers.

Still (partly) open are the concerns regarding the dependency and inherited biases from CLIP (Reviewers sbm3, E7Mu, op3v),  the capability of the proposed method at indicating “why” a model fails - it seems to only be “where” in the architecture (Reviewer E7Mu), and the actual causality of the proposed method (Reviewer 4V8N).

**Reviewer Concerns:**

Addressed Concerns:

- Reviewer sbm3

    - Compatibility with generative models and large vision-language models.

    - Missing a more detailed computational cost analysis

    - Missing user study

- Reviewer E7Mu

    - Actual interpretability of the representations taken linked from CLIP.

    - Unsupported claims.

    - Technical problem: is $E_n(a_i \cdot e^{(i)})$ OOD for the encoder.

    - The validation presented in Table 1 is insufficient; specific case of the final classification layer.

    - Superious significance report in Table 1

    - (Table 2)The way specificity of interventions is being evaluated is not correct as it currently ignores the effect of the intervention on the other classes of interest.

    - Insufficient support in Table 2,3 for Finding 2,3.

- Reviewer op3v

    - Comparison w.r.t. related works (STAIR and DeViL)

    - Including MILAN as a baseline

    - Relationship between high activation values and concepts.

    - Including CoSY as evaluation method

    - Missing baselines (Act-Value, Network Dissection and V-Interp) in Table 1.

    - Missing overall accuracy in Table 2,3.

    - Apparent low performance of the proposed method on Transformers

    - Need for the “Sig. column in Table 1.

- Reviewer 4V8N

    - Provide qualitative examples from low-level features at early layers.


Outstanding Concerns:

- Reviewer sbm3

    - Dependency on the CLIP semantic space **\- Partly**

- Reviewer E7Mu

    - Bias towards the object-centric CLIP representations. **\-** **Partly**

    - Overclaim that MICLIP can explain “why” a mode fails.

- Reviewer op3v

    - Assumption that the target layer of operation actually contains the concepts of interest. **\- Partly**

    - inherited Bias from CLIP representations. **\- Partly**

- Reviewer 4V8N

    - The proposed method does not guarantee causality.

**Reviewer Scores:**

From the four reviewers, op3v manifested openness towards increasing his/her score given that the concerns were addressed. I would expect certainly a score increase there, perhaps of around two levels (up to 8) considering the amount of his/her concerns that were addressed. I would also expect an increase from Reviewer sbm3.

In my opinion the main concern from Reviewer 4V8N is still open; I would not expect any change in his/her score considering the very positive score (8) already given by the reviewer and relatively low confidence.

Finally, Reviewer E7Mu was the most critical (initial score: 2) and while a good amount of his/her concerns were addressed, from his/her view on the paper, I believe proper discussion may not have led to a score increase of more than one level (to 4)

---

### Decision · Program_Chairs · 2026-01-26

Accept (Poster)